# Antibody-directed extracellular proximity biotinylation reveals that Contactin-1 regulates axo-axonic innervation of axon initial segments

Yuki Ogawa [1,6], Brian C. Lim[1,6], Shanu George[2], Juan A. Oses-Prieto [3], Joshua M. Rasband[1], Yael Eshed-Eisenbach[4], Hamdan Hamdan[1,5], Supna Nair[3], Francesco Boato [2], Elior Peles [4], Alma L. Burlingame [3], Linda Van Aelst [2] & Matthew N. Rasband [1] ✉

Axon initial segment (AIS) cell surface proteins mediate key biological processes in neurons including action potential initiation and axo-axonic synapse formation. However, few AIS cell surface proteins have been identified. Here, we use antibody-directed proximity biotinylation to define the cell surface proteins in close proximity to the AIS cell adhesion molecule Neurofascin. To determine the distributions of the identified proteins, we use CRISPR-mediated genome editing for insertion of epitope tags in the endogenous proteins. We identify Contactin-1 (Cntn1) as an AIS cell surface protein. Cntn1 is enriched at the AIS through interactions with Neurofascin and NrCAM. We further show that Cntn1 contributes to assembly of the AIS extracellular matrix, and regulates AIS axo-axonic innervation by inhibitory basket cells in the cerebellum and inhibitory chandelier cells in the cortex.

The axon initial segment (AIS) is essential for proper neuronal and brain circuit function. AIS integrate synaptic inputs, generate and modulate axonal action potentials, and regulate the trafficking of proteins, vesicles, and organelles to maintain neuronal polarity. These functions depend on a tightly regulated network of scaffolding and cytoskeletal proteins that serve as an organizing platform for ion channels and cell adhesion molecules (CAMs)[1,2]. However, the AIS proteins that have been described likely represent only a small fraction of the overall AIS proteome since the molecular mechanisms involved in many AIS-associated processes remain poorly defined.

Recently, proximity-dependent biotinylation (PDB) approaches have emerged as robust experimental strategies to define the molecular composition of organelles and subcellular domains[3]. PDB is particularly attractive to identify AIS proteomes since the AIS is very

detergent insoluble and refractory to more traditional proteomic approaches like immunoprecipitation (IP) mass-spectrometry. Streptavidin pulldown of biotinylated AIS proteins allows for the use of much stronger solubilizing detergents. We previously used one PDB approach (BioID) to discover new AIS proteins[4]; we targeted the biotin ligase BirA* to the AIS by fusing it to a variety of known AIS cytoskeleton-associated proteins. These experiments identified known and some new cytoplasmic AIS proteins, including Mical3 and Septins. However, our experiments were strongly biased toward cytoplasmic proteins and recovered very few membrane and cell surface proteins. Some PDB approaches have successfully captured cell surface proteins. For example, Li et al. (2020)[5] used an extracellular, membrane tethered horseradish peroxidase (HRP) to identify cell surface proteins that function as regulators of neuronal wiring; their transgenic

[1]Department of Neuroscience, Baylor College of Medicine, Houston, TX, USA. [2]Division of Neuroscience, Cold Spring Harbor Laboratory, Cold Spring Harbor, NY, USA. [3]Department of Pharmaceutical Chemistry, University of California San Francisco, San Francisco, CA, USA. [4]Department of Molecular Cell Biology, Weizmann Institute of Science, Rehovot, Israel. [5]Present address: Department of Physiology and Immunology, Khalifa University, Abu Dhabi, United Arab Emirates. [6]These authors contributed equally: Yuki Ogawa, Brian C. Lim. ✉e-mail: rasband@bcm.edu

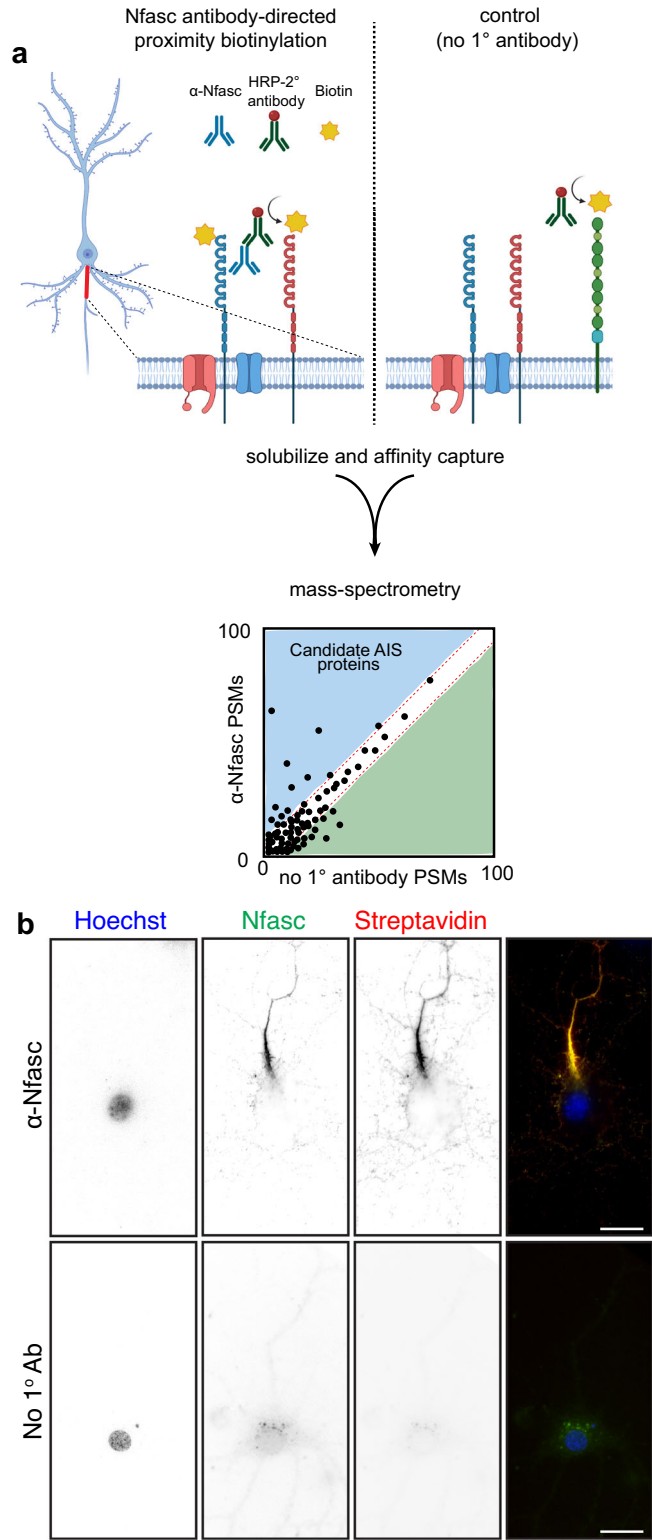

**Fig. 1 | Proximity-dependent biotinylation using Nfasc antibodies. a** Illustration of the antibody-directed proximity biotinylation strategy. Anti-Nfasc antibodies bind to Nfasc, while HRP-conjugated secondary antibodies bind to the Nfasc antibodies. Addition of biotin phenol (biotin tyramide) in an $H_2O_2$ containing diluent results HRP-mediated conversion of the biotin phenol to an active radical biotin phenoxyl that covalently adds the tyramide biotin to extracellular tyrosine residues. Omission of the primary anti-Nfasc serves as a control. After stringent solubilization and affinity capture by streptavidin-conjugated magnetic beads. Biotinylated proteins are then identified by mass spectrometry. Created with Biorender.com. **b** Fluorescence imaging of DIV14 rat hippocampal neurons labeled by Nfasc-BAR or a control condition (no primary Ab). Nfasc fluorescence (green) enrichment defines the AIS. Hoechst (blue) labels nuclei, and biotinylated proteins are detected using Alexa594-conjugated streptavidin (red). $N = 3$ independent experiments. Scale bars, 20 μm.

interactions between neuronal NrCAM and astrocytic NrCAM that stabilize the structure and function of inhibitory synapses.

To overcome some of the limitations of intracellular PDB for identification of AIS cell surface proteins, we used Selective Proteomic Proximity Labeling Assay Using Tyramide (SPPLAT)[8,9]; the approach has also been called Biotinylation by Antibody Recognition (BAR)[10]. Our application of this strategy uses highly specific primary antibodies against the extracellular domain of the AIS-enriched CAM Neurofascin (Nfasc) to direct HRP conjugated secondary antibodies to the AIS. Addition of biotin-tyramide and hydrogen peroxide generates biotin phenoxyl radicals that biotinylate membrane proteins within a range of ~250 nanometers of the peroxidase[11]. We performed this labeling at multiple timepoints throughout neuronal development in vitro on live neurons. We identified all previously reported AIS extracellular, and membrane cell adhesion and recognition molecules. In addition, we found many membrane proteins that were reproducibly in proximity to Nfasc, with different temporal enrichment profiles. We further investigated a subset of these using CRISPR-mediated endogenous gene tagging. Among these, we identified Contactin-1 (Cntn1) as a previously unidentified, bona fide AIS CAM recruited to the AIS through interaction with the AIS CAMs Nfasc and NrCAM. We found that loss of Cntn1 severely impaired inhibitory axo-axonic innervation of the AIS in both cerebellar Purkinje neurons and cortical pyramidal neurons. Thus, using antibody directed extracellular proximity biotinylation, we identified Cntn1 as an AIS protein that regulates axo-axonic innervation of the AIS.

## Results

### Proximity biotinylation at the AIS membrane

We reasoned the Nfasc proximity proteome could be used to help define the AIS cell surface proteome since Nfasc is highly enriched at the AIS. Therefore, we adapted the SPPLAT/BAR method[8,10] for use with live, unpermeabilized neurons; we avoided fixation to maximize protein recovery and subsequent mass spectrometry. We labeled rat hippocampal neurons in culture with highly specific and validated chicken primary antibodies against the ectodomain of Nfasc[12], since its 186 kDa isoform (NF186) is highly enriched at the AIS (Fig. 1a, b), with lower concentrations along the distal axon, at growth cones, and in the soma[13]. After live labeling with the anti-Nfasc primary antibody, HRP-conjugated anti-chicken secondary antibodies were used to label the anti-Nfasc primary antibody. The Nfasc-localized HRP generates the reactive biotin phenoxyl from biotin tyramide (biotin phenol), resulting in the biotinylation of tyrosine residues in proximity to Nfasc with a range of several hundred nm[9,11]. As with other PDB methods, non-specifically and endogenously biotinylated proteins, as well as non-specific protein background adsorbing to solid phase surfaces during the enrichment steps and prior to the mass spectrometry analysis, must be excluded. The omission of the primary antibody serves as a simple and straightforward negative control. Without fixation or

approach revealed the cell surface proteome of Drosophila olfactory projection neurons. Shuster et al. [6]. used the same approach in mice, but restricted the expression of the membrane tethered HRP to Purkinje neurons to reveal their cell surface proteome. However, neither of these approaches was designed to interrogate subcellular domains. As an alternative approach, Takano et al. (2020)[7] used a split PDB strategy (Split-TurboID) to elucidate the cell surface proteome of astrocyte-neuron synapses. Their experiments revealed transcellular

detergents, the membrane-impermeability of biotin-phenoxyl restricts the biotinylation reaction to the extracellular surface. Hereafter, we refer to this method as Nfasc-BAR.

We found that Nfasc-BAR resulted in AIS-enriched streptavidin labeling that colocalized with NF186 (Fig. 1b); this pattern was not seen when the anti-Nfasc antibody was omitted. The amount of biotinylation also depends on the duration of the labeling reaction (Fig. S1a). For the experiments described below, we used a reaction time of 5 min (Fig. S1b). To identify biotinylated AIS proteins, we then solubilized neuronal membranes using a strong solubilization buffer, purified biotinylated proteins using streptavidin-conjugated magnetic beads, and finally identified the biotinylated proteins using mass spectrometry (Fig. 1a). To confirm the reproducibility and robustness of our approach we performed surface proximity labeling in parallel using rabbit polyclonal antibodies targeting the ectodomain of NrCAM (Fig. S1c), another AnkG-binding CAM found at the AIS. Proximity biotinylation directed by NrCAM antibodies strongly labeled the AIS (Fig. S1c). Importantly, the resulting mass spectrometry datasets confirmed the robustness of the strategy since Nfasc-BAR and NrCAM-BAR proximity proteomes were highly concordant (Fig. S1d; supplementary Data file 1).

## NF186 proximity proteomes across neuronal development
The maturation of axons includes the enrichment of proteins that mediate key functions or developmental mechanisms. For example, toward the end of the first week in vitro, the scaffolding protein AnkyrinG (AnkG) localizes to the proximal axon; this enrichment precedes and is necessary for the subsequent recruitment of Nfasc, and voltage-gated $Na^+$ (Nav) and $K^+$ (Kv) channels to the AIS[14,15]. NF186 enrichment at the AIS and along distal axons also increases during development (Fig. 2a). To determine the extracellular Nfasc proximity proteome and how it changes during development (both before and after AIS formation), we performed Nfasc-BAR on primary hippocampal neurons at five different timepoints (Supplementary Data file 2): from day in vitro 4 (DIV4; prior to AIS formation) to DIV28 after establishment of the AIS-associated ECM (Fig. S2). To compare Nfasc cell surface proximity proteomes from cultures of different ages, we normalized peptide spectral match (PSM) counts to total spectral counts in the set of endogenously biotinylated carboxylases detected (Fig. S2), as a correction factor for differences in total protein amount used in individual pulldowns (see methods). We found that as neurons, the AIS, and axons develop, the cell surface Nfasc proximity-proteome changes, with an increasing number of proteins displaying significant changes in fold enrichment (Fig. 2, Supplementary Data file 2). Consistent with a developing and maturing AIS and increasing levels of overall Nfasc, volcano plots (Fig. 2b–f and Supplementary Data file 3) show the enrichment of proteins identified using Nfasc-BAR compared to controls at the various developmental time points. We used a cutoff of $log_2$(Nfasc PSMs/Ctrl PSMs) or $log_2$[fold change (FC)] >2 (vertical dotted line) with a significance cutoff of $p < 0.05$ (horizontal dotted line).

To visualize the increase in proteins identified using Nfasc-BAR across development and to select candidates to focus on, we identified 285 proteins that satisfied two filtering criteria for at least one of the five timepoints: (1) normalized PSMs > 10 and (2) $log_2$(Nfasc PSMs/Ctrl PSMs) or $log_2$(FC) > 2 (Fig. S2; Supplementary Data file 2). Among these 285 proteins there were a variety of protein expression profiles (Fig. 3a). Although present, relatively few proteins showed a reduction in the cell surface Nfasc proximity proteome. Most proteins identified in the Nfasc proximity proteome increased in abundance (Fig. 3b; only the 100 candidates with the largest increase across all time points are shown). We also plotted the $log_2$FC at each time point for the 100 proteins with the largest fold change (DIV 4 only had 63 proteins with $log_2$FC > 2) (Fig. S3). The results for each protein were highly reproducible at each time point and consistently revealed similar sets of cell

surface proteins. Cytoplasmic AIS proteins such as AnkG, β4 spectrin, and TRIM46 were conspicuously absent consistent with our experimental design to restrict the biotinylation to cell surface proteins. In addition, gene ontology (GO) pathway analysis was consistent with the identification of membrane proteins whose main functions include adhesion, receptor, and extracellular matrix binding (Fig. S4).

## The Nfasc-BAR proximity proteome includes AIS enriched proteins
Among the cell surface proteins that passed our selection criteria (Fig. S2), we found all known AIS membrane proteins and AIS enriched extracellular matrix molecules with the notable exception of ion channels (Fig. 4a). Why might that be the case since ion channels are known to be highly enriched at the AIS? The number of PSMs recovered for any protein depends on: (1) the number of available extracellular tyrosine residues (Nfasc-BAR-mediated biotinylation occurs on tyrosine residues); (2) the amount and local membrane density of the protein; and (3) the proximity of the protein to the biotinylation source. Since ion channels are highly enriched at the AIS and are in close proximity to Nfasc, their absence from our data set may reflect the small number of extracellular tyrosine residues found in ion channels and a topology that has extracellular residues very close to the membrane. For example, the AIS-enriched $K^+$ channel subunit KCNQ3 has four extracellular regions comprising 45 amino acids, with two of those being tyrosine (Fig. 3b); we did not detect any KCNQ3 peptides in our experiments. Similarly, Nav1.2 (encoded by *Scn2a*), the main voltage-gated $Na^+$ channel expressed at AIS in developing neurons[16], has an average of one tyrosine per extracellular domain and many are very close to or immediately adjacent to the membrane. In contrast to ion channels, cell adhesion molecules like Nfasc and Contactin-1 (Cntn1) have large extracellular domains with many tyrosine residues (Fig. 4b). Thus, the low number of extracellular tyrosines found in ion channels and the proximity of these residues to the membrane may make them difficult to biotinylate using Nfasc-BAR; other membrane or cell surface proteins with few or inaccessible tyrosine residues may also be poorly represented in our data set.

In contrast to ion channels, an analysis of the 201 proteins identified at DIV14 showed no correlation between the number of PSMs for a protein and the number of extracellular tyrosines (Fig. 4c, d). This suggests a much stronger dependence of PSM number on proximity and protein abundance. As an estimate for both proximity and abundance, we calculated the ratio of extracellular tyrosines to PSM count for all 201 proteins identified at DIV14. Thus, a lower ratio suggests greater abundance of protein and closer proximity to the HRP-dependent biotinylation source (Fig. 4e). This analysis shows many candidates with low extracellular tyrosine/PSM ratios that were also previously reported to be directly or indirectly linked to Nfasc, including PlxnA4, Ncam1, L1CAM, and NrCAM (Fig. 4e)[17,18].

## Tagging of endogenous membrane proteins
Our results across 5 developmental timepoints yielded an NF186 proximity proteome (Fig. 2a); filtering based on fold-enrichment and number of PSMs recovered resulted in 285 candidate cell surface proteins in close proximity to NF186 (Fig. S2). Among these, Nfasc-BAR successfully identified the 13 previously reported AIS-enriched cell surface proteins (excluding ion channels; Fig. 5a). However, it is unlikely that the remaining 272 proteins are also enriched at the AIS since NF186 is present in lower densities in the soma, axons, and at growth cones[13], some of the proteins were identified before the AIS forms or is mature (e.g., DIV4), and the range of SPPLAT/BAR is ~200–300 nm[9,11]. Thus, proteins identified by Nfasc-BAR may be in proximity to NF186 but not enriched at the AIS. We previously used antibodies to validate the AIS proteomes we identified using BioID[4]. However, antibodies are frequently non-specific and for reasons that are unclear, many antibodies that label AIS are not against their

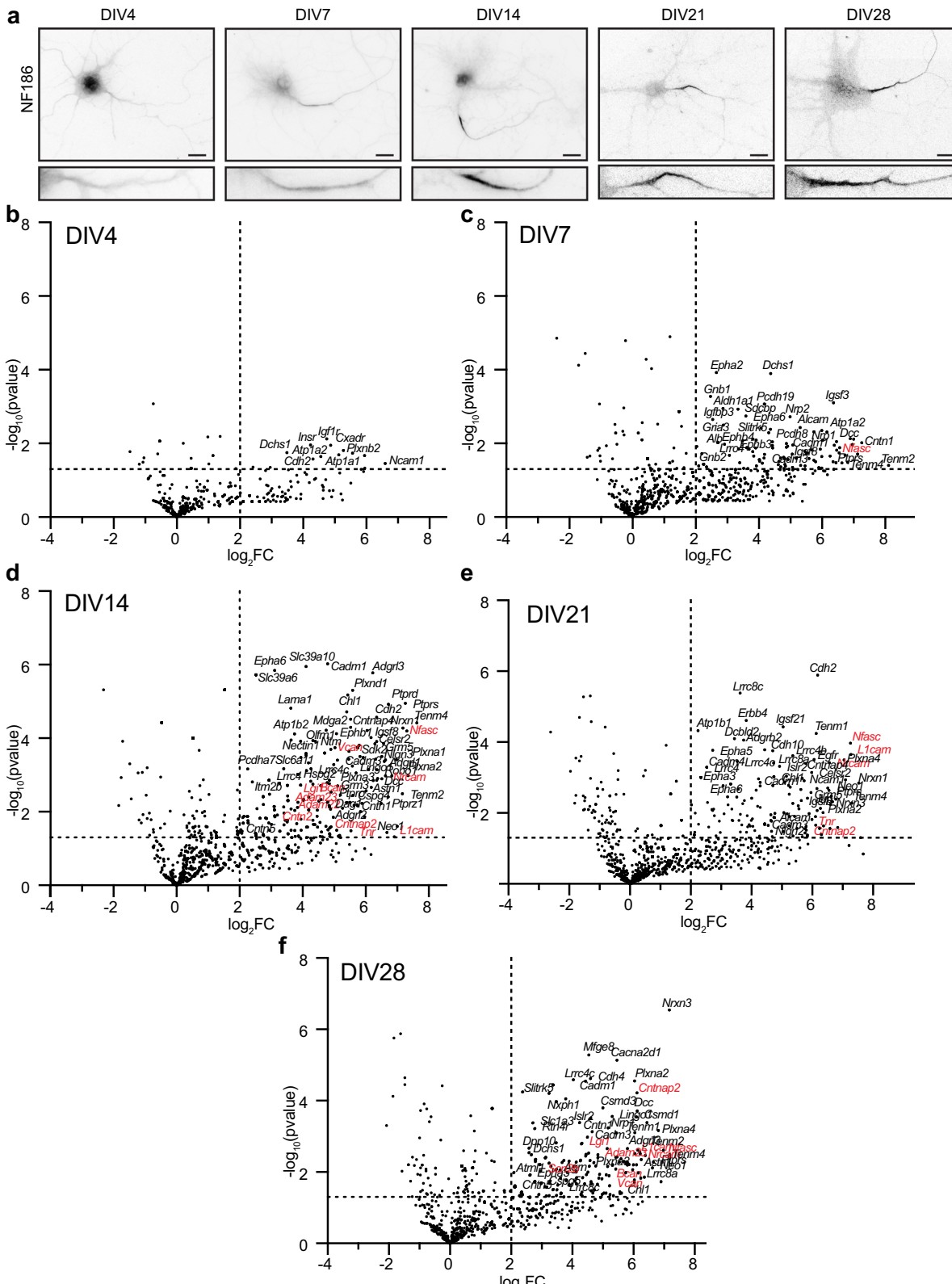

**Fig. 2 | NF186 proximity proteomes across neuronal development.**
**a** Immunofluorescence labeling of NF186 at different timepoints of hippocampal neuron development in vitro; $N = 3$ independent experiments. Lower panels: magnified images show the Nfasc-labeled AIS at each time point. Scale bars, 20 μm. **b**–**f** Volcano plots showing the $\log_2$-fold changes of proteins versus the statistical significance $-\log_{10}$(pvalue) identified using Nfasc-directed proximity biotinylation

($N = 3$). P values were calculated using a nonparametric two-sided T-test; no adjustments were made for multiple comparisons. $P < 0.05$ was used as a cutoff for significance (horizontal dashed line). Some identified proteins are indicated (corresponding gene names listed); those previously reported as AIS cell surface proteins are shown in red.

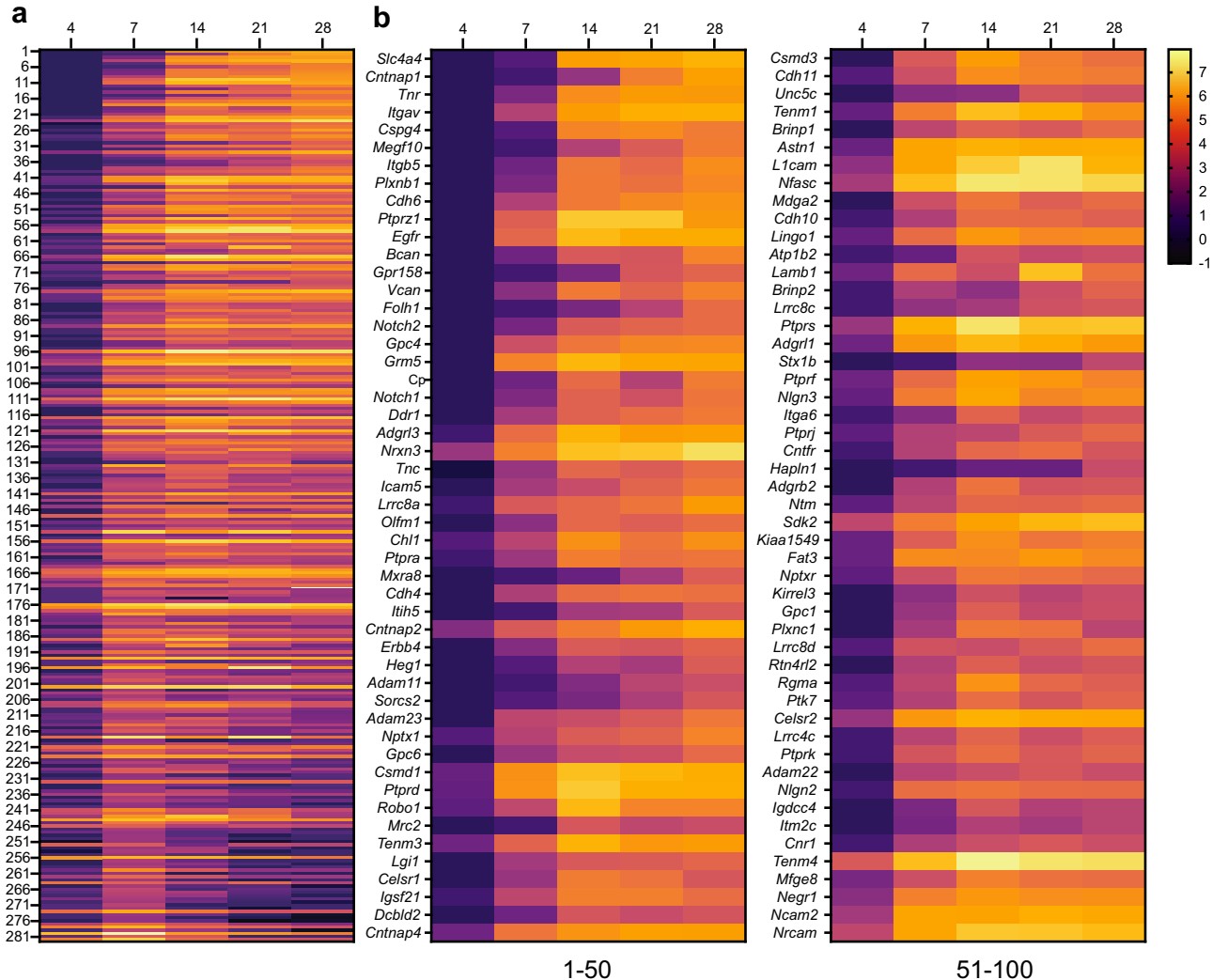

**Fig. 3 | Change in expression level for cell surface proteins in proximity to Nfasc. a** Heatmaps showing log$_2$-fold changes at each timepoint for all 285 proteins that satisfied two filtering criteria [(1) normalized PSMs > 10; (2) log$_2$FC (Nfasc/Ctrl) > 2] for at least one of five timepoints, rank-ordered by the slope of the linear regression of their log$_2$ fold enrichment over time. **b** Expanded heat map showing gene names for the proteins (1–50 and 51–100) with the largest rate of increase in PSM count (B). Data shown are from $N = 3$ biological replicates for each timepoint (see Fig. S2).

claimed targets[19–21]. Therefore, to circumvent some of the challenges associated with antibodies and to look for AIS- and axon-enriched proteins, we performed CRISPR-mediated epitope tagging of endogenous proteins[20,22–24]. We selected 23 different candidates (Fig. 5a) identified using Nfasc-BAR based on (1) the high fold-enrichment compared to control BAR, (2) the high number of PSMs recovered, and (3) the estimate of proximity to the biotin source (Figs. 2 and 4). Included in these 23 candidates were four cell adhesion molecules previously reported at the AIS: Nfasc, NrCAM, L1CAM, and Cntn2[25–27]. To endogenously label these 23 cell surface proteins, we generated two adeno-associated viruses (AAV) to transduce cultured DIV 0 rat hippocampal neurons with (1) Cas9 and (2) a gene specific single guide RNA (sgRNA), a sgRNA that recognizes donor recognition sites (DRS) flanking spaghetti monster fluorescent protein with V5 tags (smFP-V5), and smFP-V5 (Fig. 5b). The gene specific sgRNAs were targeted to the last exon of each gene of interest allowing for the insertion of smFP-V5 in the last exon. Two weeks after transduction, neurons were fixed and immunostained for β4 spectrin to label the AIS, and V5 to detect the endogenously tagged cell surface protein. Since we targeted the last exon (C-terminus) of each protein resulting in premature termination of the protein, it is possible the addition of the smFP-V5 disrupted the normal localization of the cell surface protein. However, C-terminal tagging of endogenous Nfasc, NrCAM, L1CAM, and Cntn2 all resulted in AIS labeling as previously reported[26–28] (Fig. 5c–f). The candidates we tested labeled AIS, axons, and dendrites (Figs. 5 and S5). For example, endogenous tagging of Ncam1 revealed uniform surface labeling in somatodendritic, AIS, and axonal domains (Fig. 5g), while Ptprs and Tenm4 showed preferential labeling of AIS and axons (Fig. 5h, i); endogenous labeling of Adgrl3 strongly labeled dendrites and spines (Fig. 5j). Among all the candidates we tested that had not previously been reported at the AIS, we found that endogenous tagging of Cntn1 showed the strongest labeling at the AIS (Fig. 5k).

## Cntn1 is a bona fide AIS cell surface protein

Cntn1 is a glycosylphosphatidyl inositol (GPI)-anchored cell adhesion molecule widely expressed throughout the nervous system in both neurons and glia[29]. It has essential roles in forming the axoglial junctions flanking nodes of Ranvier where it forms a complex together with axonal Caspr and the glial 155 KDa splice variant of Nfasc (NF155)[30,31]. Cntn1-null mice die in the 3rd postnatal week, emphasizing the importance of Cntn1 to normal function. Cntn1 was also reported at nodes of Ranvier, although its function there is unknown[32]; detection of nodal or paranodal Cntn1 requires different fixation and treatment

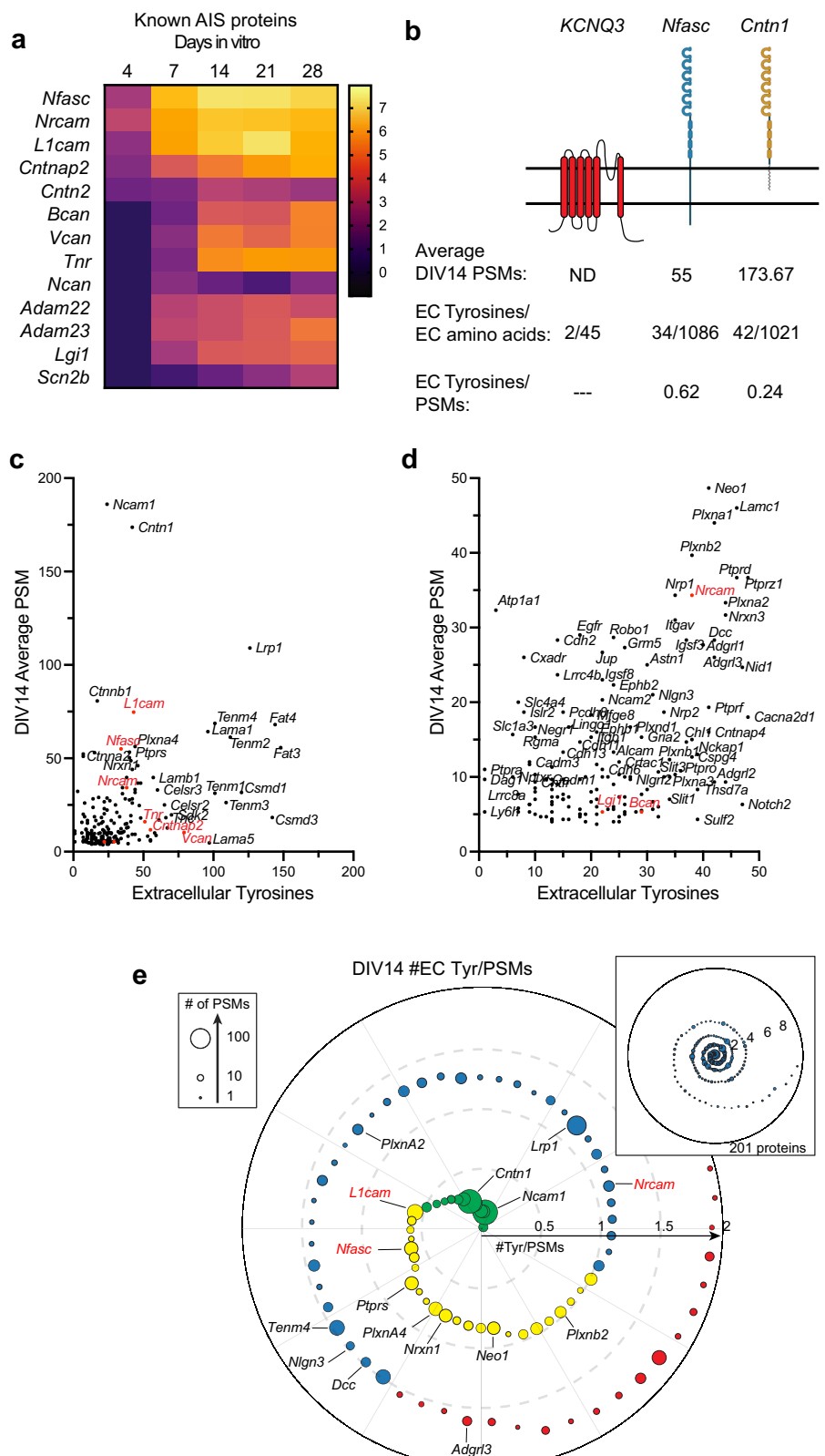

**Fig. 4 | Nfasc-BAR identifies known AIS membrane and membrane-associated proteins. a** Known AIS membrane and membrane-associated proteins and their log$_2$FC (Nfasc/Ctrl). **b** Illustration of membrane topology, average PSMs (at DIV14), and the number of extracellular tyrosine residues for three different AIS and membrane proteins. Created with Biorender.com. **c**, **d** Scatter plot of the number of peptide spectral matches (PSMs) for each biotinylated protein identified by mass spectrometry as compared to the number of tyrosine residues present in each protein's extracellular domain, shown at different scales. Proteins in red were previously reported at the AIS. **e** Proximity plot showing biotinylated proteins (at DIV14) ordered by extracellular (EC) tyrosine/PSM ratio. The plot is an estimate of abundance and proximity to the HRP secondary antibody bound to the Nfasc primary antibody. Each protein is represented by a circle with size proportional to the number of PSMs identified for that protein. Proteins analyzed in subsequent experiments are indicated by their gene names. EC Tyrosine/PSM ratio: 0–0.5 green, 0.5–1.0 yellow, 1.0–1.5 blue, 1.5–2.0 red.

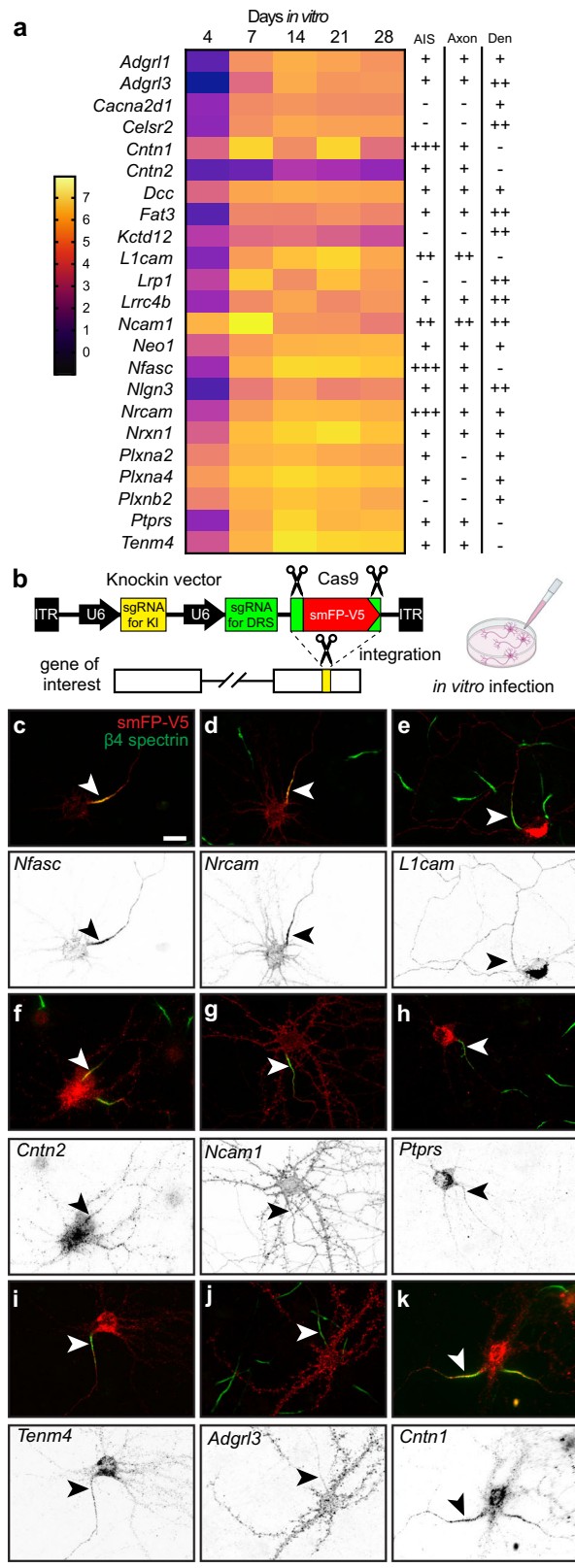

**Fig. 5 | Tagging of endogenous membrane proteins. a** Proteins whose distribution was tested using endogenous protein tagging. The heatmap shows the increase in expression level is shown as a function of days in vitro (see also Fig. 2c). The presence of the tagged protein in AIS, axon, and dendrite is indicated. **b** Schematic of the knock-in vector for in vitro CRISPR-mediated endogenous gene tagging. DRS, donor recognition sites. Created with Biorender.com. **c–k** Examples of smFP-V5 tagged proteins (red) enriched at the AIS (**c, d, f, k**), the axon (**e, g**), dendrites (**j**), or in multiple domains (**e, g, h, i**). AIS are labeled for β4 spectrin (green) and are indicated by an arrowhead. Neurons were transfected at DIV0 and fixed at DIV21. $N = 2$ independent experiments with two independent viruses targeting each gene of interest. Scale bar, 20 μm.

control or three *Cntn1* specific sgRNAs (Fig. 6a). Whereas neurons transduced with Cas9 and the control sgRNAs had Cntn1 and AnkG immunolabeling at the AIS (Fig. 6b, arrowheads), neurons transduced with the *Cntn1* sgRNAs lost both the perisomatic and AIS Cntn1 immunoreactivity (Fig. 6c, arrowhead; 6d). Thus, immunostaining of cultured hippocampal neurons reveals AIS Cntn1. These results also demonstrate the specificity of the Cntn1 antibody. However, loss of AIS Cntn1 had no effect on AIS AnkG (Fig. 6c, arrowhead), Nfasc, Na⁺ channels (Fig. S6a, b), or the ratio of AIS Nfasc/β4 spectrin levels (Fig. S6c).

Transduction of DIV10 cultured hippocampal neurons using AAV to express myc-tagged Cntn1 (Fig. 6e, f) also showed Cntn1-myc enriched at the AIS that colocalized with β4 spectrin at DIV14 (Fig. 6f, arrowheads; Fig. 6h). Similarly, retro-orbital injection of AAV Cntn1-myc in 13-week old mice, showed strong AIS enrichment of Cntn1-myc in transduced cortical neurons 4 weeks after injection (Fig. 6e, g, arrowheads; Fig. 6h). Finally, we performed in vivo AAV-dependent and CRISPR-mediated tagging of endogenous Cntn1 using smFP-V5 (Fig. 6i). As with cultured neurons (Fig. 5k), we found the sgRNA targeting *Cntn1* resulted in V5 labeling of cortical neuron AISs (Fig. 6j, arrowheads; Fig. 6k). Together, these results show that Cntn1 is a bona fide AIS cell surface protein.

### Cntn1 is localized at the AIS through binding to L1-family cell adhesion molecules

Since Cntn1 is a GPI-anchored cell surface protein, we reasoned that it must be recruited to the AIS through interactions with a co-receptor or some other AIS transmembrane protein. Cntn1 has 6 N-terminal immunoglobulin-like (Ig-like) and 4 C-terminal fibronectin III (FNIII) domains (Fig. 7a). To determine how Cntn1 is clustered at the AIS, we generated myc-tagged Cntn1 with various internal deletions of these domains. We found the N-terminus and first four Ig-like domains of Cntn1 are required for its AIS clustering (Fig. 7b, d). In contrast, deletion of the last two Ig domains or any of the FNIII domains did not affect recruitment of Cntn1 to the AIS (Fig. 7c, d).

What membrane protein recruits Cntn1 to the AIS? Biochemical and cell biological studies suggest that Cntn1 interacts with members of the AnkG-binding L1 family of cell adhesion molecules including Nfasc, L1CAM, and NrCAM[34,35]. Although all three of these CAMs are enriched at the AIS, L1CAM is also found at high levels along the distal axon (Fig. 5c–e). To determine if the AIS-enriched Nfasc or NrCAM recruits Cntn1 to the AIS, we generated sgRNAs vectors (Fig. 6a) to delete Nfasc and NrCAM from neurons. Surprisingly, removal of Nfasc or NrCAM alone had no effect on the clustering of Cntn1 at the AIS (Fig. 7e, f). However, simultaneous deletion of both Nfasc and NrCAM blocked the AIS clustering of Cntn1 (Fig. 7e, arrowhead; f). To further define the mechanism of Cntn1 targeting to the AIS, we introduced 3 point mutations found in Ig1-Ig4 that disrupt the interaction between Nfasc and Cntn1[36]. Whereas control Cntn1 was highly enriched at the AIS (Fig. S7a, c), the mutant Cntn1 (Cntn1-mut) failed to be enriched at the AIS (Fig. S7b, c). Together, these results suggest that the AIS enriched CAMs Nfasc and NrCAM redundantly recruit Cntn1 to the AIS through Ig1-Ig4.

conditions[33], suggesting that in some subcellular domains Cntn1 may engage in protein-protein interactions that preclude immunostaining. With this in mind, our efforts to immunolabel Cntn1 at AIS in control mouse brain failed. Nevertheless, we performed immunostaining in vitro using a goat-polyclonal anti-Cntn1 antibody in control and Cntn1-deficient neurons. We disrupted endogenous Cntn1 expression in cultured hippocampal neurons using AAV to express Cas9 and three

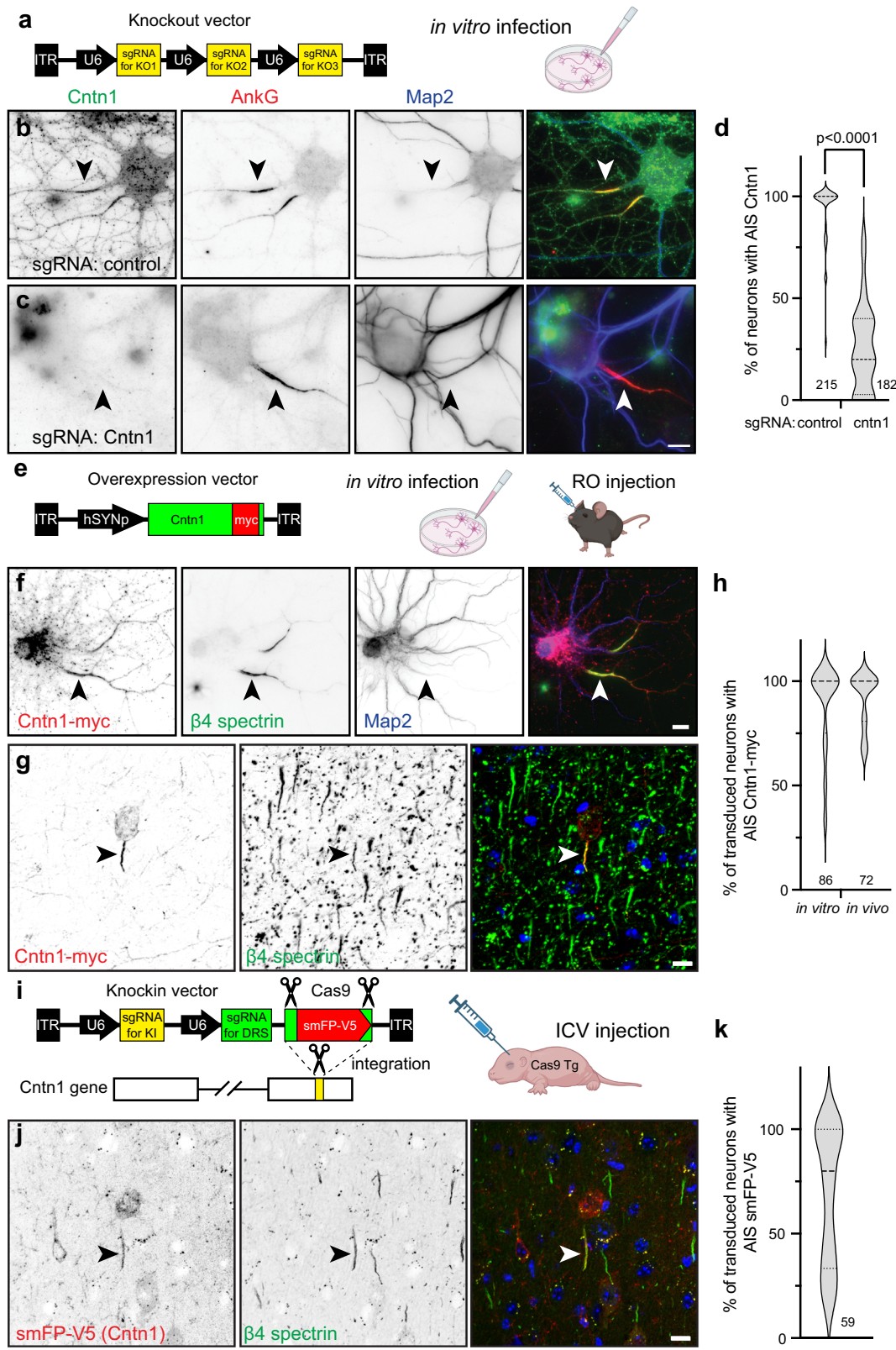

## Cntn1 helps assemble the AIS extracellular matrix

We found Tenascin-R (Tnr) in our Nfasc-BAR proximity proteome (Fig. 4a). Tnr is an extracellular matrix molecule and a known Cntn1 interactor[34]. Immunostaining of cultured hippocampal neurons using antibodies against Tnr showed that it strongly labeled the AIS and colocalizes with Nfasc (Fig. 7g). Previous studies showed that Nfasc regulates the AIS recruitment of the extracellular matrix molecule

Brevican (Bcan)[15]. To determine if Nfasc or Cntn1 also regulate the AIS clustering of extracellular Tnr and Bcan, we disrupted expression of *Nfasc* or *Cntn1*; sgRNA vectors (Fig. 6a) targeting *Nfasc* and *Cntn1* were highly efficient (Figs. 6c and 7h). Although control sgRNAs had no effect on AIS Tnr or Bcan, loss of Nfasc or Cntn1 significantly reduced Tnr's AIS enrichment (Fig. 7h, i), and loss of Cntn1 significantly disrupted the AIS clustering of Bcan (Fig. S6d). Thus, Nfasc and Cntn1

**Fig. 6 | Cntn1 is an AIS protein. a** The knockout vector for in vitro transduction of neurons. **b, c** Immunostaining for Cntn1 (green), AnkG (red) and Map2 (blue) after transduction with control (**b**) or Cntn1 (**c**) sgRNAs; arrowheads indicate AIS. Neurons were transduced at DIV0 and fixed at DIV21. Scale bar, 10 μm. $N = 8$ independent experiments. **d** Quantification of Cntn1 knockout by sgRNA. $N = 8$ coverslips, 4 FOV for each coverslip and each sgRNA; $n = 215$ and 182 neurons for control and Cntn1 sgRNAs, respectively. P values calculated using a nonparametric two-sided T-test; no adjustments were made for multiple comparisons; ($t = 14.07$, df = 62) $p = 5.79 \times 10^{-21}$. **e** The Cntn1-myc vector used for infection of neurons. RO, retro-orbital. **f** Transduction of hippocampal neurons at DIV 10 using AAV to express myc-tagged Cntn1. Neurons were fixed at DIV14. Cntn1-myc (red) is enriched at the AIS (arrowhead) and colocalizes with β4 spectrin (green). Map2 (blue) labels somatodendritic domains. $N = 3$ independent experiments. Scale bar, 10 μm. **g** In vivo transduction of cortical neurons in 13-week-old mice to express myc-tagged Cntn1. Brains were collected 4 weeks later. Cntn1-myc (red) is enriched at the AIS (arrowhead) and colocalizes with β4 spectrin (green); Hoechst (blue) labels nuclei. $N = 4$ mice. Scale bar, 10 μm. **h** The percentage of transduced neurons with AIS Cntn1-myc. For in vitro experiments, $N = 3$, with 15 fields of view (FOV) and 86 transduced neurons. For in vivo experiments, $N = 4$, with 16 FOV and 72 transduced neurons. **i** The knock-in vector for in vivo tagging of Cntn1. DRS, donor recognition sites. AAV were delivered by intracerebroventricular (ICV) injection at P0 into Cas9 transgenic (Tg) mice. **j** In vivo transduction of P0 cortical neurons to tag endogenous Cntn1 (red). Brains were collected 8 weeks later. The smFP-V5 tagged Cntn1 colocalizes with AIS β4 spectrin (green, arrowhead). Hoechst dye (blue) labels nuclei. $N = 4$ mice. Scale bar, 10 μm. **k** The percentage of cortical neurons with AIS smFP-V5 labeling. $N = 4$ mice, with 15 FOV and 59 transduced neurons. In all violin plots dotted and dashed lines indicate quartiles and median, respectively. **a, e, i** Created with Biorender.com.

both contribute to the assembly or stabilization of the Tnr and Bcan-containing AIS extracellular matrix.

## Cntn1 regulates the assembly of pinceau synapses in the cerebellum

Purkinje neuron AISs are innervated by inhibitory basket cell interneurons that powerfully modulate neuronal excitability[37]. Basket cells form a stereotypical 'pinceau' synapse (Fig. 8a), with presynaptic terminals at the AIS highly enriched in Kv1 K$^+$ channels and PSD-95, among other proteins[38]. The loss of the AIS scaffolding protein AnkG disrupts pinceau synapses and the AIS clustering of Nfasc, suggesting that their assembly requires AnkG-dependent clustering of CAMs like Nfasc and NrCAM[39]. Since AIS enrichment of Cntn1 also depends on these CAMs, we wondered if Cntn1 plays important roles in cerebellar pinceau synapse assembly. Therefore, we examined pinceau synapse formation in P18 Cntn1$^{-/-}$ mice; Cntn1$^{-/-}$ mice are very sick and typically die before 3 weeks of age[31]. Immunostaining of Purkinje neurons using antibodies against Nfasc, Kv1.2, and PSD95 showed stereotypical enrichment and clustering of these proteins at the AIS of control heterozygote Cntn1$^{-/+}$ mice, but Cntn1$^{-/-}$ mice had profoundly disrupted pinceau synapse formation (Fig. 8b, arrows) with significantly reduced Kv1.2 and PSD95 intensity at the pinceau synapse (Fig. 8c). Thus, Cntn1 is required for proper assembly of pinceau synapses at the AIS of cerebellar Purkinje neurons.

## Cntn1 regulates axo-axonic innervation of pyramidal neurons by chandelier cells

In the cortex and hippocampus, axo-axonic synapses are formed between chandelier cells (ChCs, also known as axo-axonic cells) and the AIS of pyramidal neurons (PyNs). ChCs are derived from progenitors in the ventral region of the medial ganglionic eminence. They have a unique axonal arbor consisting of multiple arrays of short, vertically oriented terminals of presynaptic boutons called cartridges, and each of these cartridges selectively innervates neighboring PyN AISs. These ChCs powerfully reduce PyN output by inhibiting AIS excitability; this inhibition can subsequently modulate brain circuit function and behavior[40–42]. However, the mechanisms that control the precise innervation of PyN AISs by ChCs remain incompletely understood, with ankyrin-interacting L1CAM so far being the only CAM known to be required for ChC/PyN AIS innervation[43]; since L1CAM is found throughout the axon and not just at the AIS, additional mechanisms must exist to allow for precise AIS innervation. To determine if Cntn1 regulates ChC/PyN AIS innervation, we performed in utero electroporation (IUE) using control or Cntn1-targeting sgRNA- and smFP-HA-expressing plasmids in Nkx2.1-CreER;Rosa26-loxp-STOP-loxp-tdTomato (Ai9) pregnant mice at embryonic day 15.5 (E15.5) (Fig. 9a). This timing of IUE results in disruption of the Cntn1 gene in layer II/III PyNs. At E18.5 tamoxifen was administered to the pregnant mother to induce expression of tdTomato red fluorescence protein (RFP) in a sparse group of layer II ChCs (Fig. 9b). We collected brains from P17 mice and analyzed the innervation and assembly of inhibitory synapses on PyN AISs by immunostaining for AnkG or β4 spectrin to label AISs, and antibodies to gephyrin and VGAT to label post- and pre-synaptic compartments of GABAergic synapses, respectively. We found that transfection with Cntn1 sgRNA significantly reduced the percentage of PyN AISs innervated by single RFP-positive ChCs in layer II/III of the somatosensory cortex, as compared to control sgRNA (Fig. 9c, arrowheads, and d). We found that Cntn1-deficient neurons also had significantly fewer inhibitory synapses along their AIS as indicated by gephyrin (Fig. 9e and f) and VGAT puncta (Fig. 9g and h). In contrast, we found no change in AIS length (Fig. S8a), β4 spectrin fluorescence intensity (Fig. S8b), or in the distance of the AIS from the cell body (Fig. S8c). Thus, loss of Cntn1 did not affect AIS structure or position. Together, these results show that Cntn1 is required for efficient ChC/PyN AIS innervation and consequently, the proper assembly of AIS axo-axonic inhibitory synapses.

## Discussion

AIS properties essential for brain function include: (1) high densities of ion channels, (2) mechanisms to regulate neuronal polarity, and (3) precise innervation by inhibitory interneurons. The molecular mechanisms regulating these properties all converge on the AIS scaffolding protein AnkG[39,43–45]. However, the distinct proximal mechanisms regulating these AIS properties remain poorly understood, highlighting the need to define the composition of the AIS in much greater detail. BioID-dependent cytoplasmic proximity biotinylation and differential mass spectrometry have partially elucidated AIS proteomes[4,46], but they are clearly deficient in proteins involved in transient interactions, posttranslational modifications, and cell surface proteins mediating extra- and intercellular interactions. Thus, experimental approaches that address these deficiencies are needed.

We aimed to use Nfasc-BAR to identify AIS cell surface proteins. The results were highly reproducible at each developmental time point in vitro, but showed changing profiles of cell surface proteins during development. As expected, our data sets included known AIS cell surface proteins, and consisted almost exclusively of cell surface proteins whose expression levels increased with neuronal maturation. The strategy is highly flexible and can be used with other AIS specific antibodies including those targeting cytoplasmic epitopes after detergent solubilization; the strategy can also be applied to other neuronal compartments including synapses, dendrites, and growth cones so long as highly specific and validated antibodies are used.

Since Nfasc is highly enriched at the AIS, we expected AIS-enriched membrane proteins to be over-represented in our data set. However, Nfasc is also found at lower densities in somatodendritic and distal axonal domains, and their total membrane area exceeds that of the AIS. Thus, despite the high density of AIS Nfasc, the total pool of Nfasc in non-AIS membrane is likely much greater. HRP-mediated biotinylation is very efficient and has a range ~25 times greater than BioID or APEX[11]. Thus, the proteins we recovered are more accurately

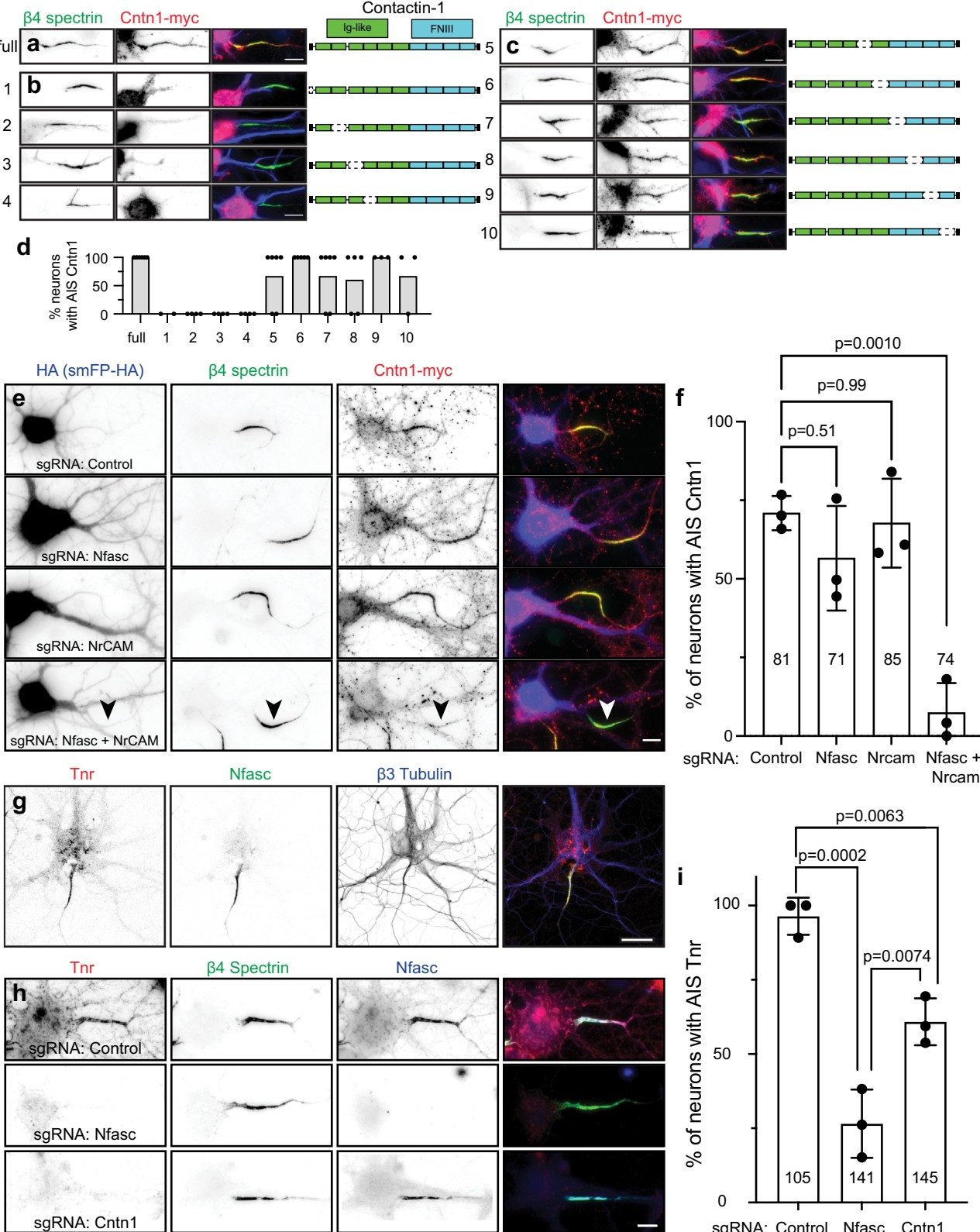

described as an Nfasc surface proximity proteome, with a subset of those proteins also being found at the AIS.

Given the promiscuity of Nfasc-BAR, the biggest challenge in our experiments was to determine which proteins to focus on and how to validate their presence or enrichment at the AIS. To this end, we narrowed our analysis using stringent filtering criteria including fold-enrichment, significance of that enrichment, and a minimum number of PSMs recovered. We also estimated the relative proximity to Nfasc based on the ratio of extracellular tyrosine residues to the PSMs recovered. Nfasc-BAR and the filtering criteria used here may underestimate proximity or miss AIS proteins that were not well biotinylated (e.g., ion channels). Nevertheless, among the proteins recovered and that satisfied the filtering criteria, our use of endogenous gene tagging revealed several that were present or even enriched at the AIS, but that

**Fig. 7 | Cntn1 is recruited to the AIS through interactions with AnkG-binding L1-family cell adhesion molecules. a** Cntn1-myc (red) colocalizes with β4 spectrin (green). Cntn1 consists of 6 N-terminal Immunoglobulin (Ig)-like domains and 4 C-terminal Fibronectin type III (FNIII) domains. Scale bar, 10 μm. **b** Cntn1-myc with N-terminal and internal deletions of the first 4 Ig-like domains. Scale bar, 10 μm. **c** Cntn1-myc localization to the AIS does not depend on the last 2 Ig-like domains or any FNIII domain. Scale bar, 10 μm. For experiments in (**b** and **c**), neurons were transfected at DIV15 and fixed one day later at DIV16. **d** The percentage of neurons with Cntn1-myc (full) or Cntn1-myc truncation variants (variants 1-10) at the AIS. $N = 2, 4, 4, 4, 6, 5, 6, 5, 3$, and 3 FOV for constructs 1–10, respectively; individual data points and mean are shown. Data are from two independent experiments. **e** Hippocampal neurons transduced with AAV to express Cas9 and control, Nfasc, NrCAM, or Nfasc+NrCAM gRNAs at DIV0 (these AAVs also express smFP-HA). At DIV12 neurons were transduced with Cntn1-myc AAV, and fixed at DIV21. Neurons

were labeled using antibodies against HA as a transduction marker (blue), β4 spectrin (green), and Cntn1-myc (red). Scale bar, 10 μm. **f** Quantification of the percentage of transduced neurons with AIS Cntn1-myc. $N = 3$ independent experiments, 5-13 FOV per experiment. Ordinary one-way ANOVA with Tukey's multiple comparisons test. Error bars, ±SEM. The total number of neurons analyzed is also indicated. **g** Immunostaining of cultured hippocampal neurons using antibodies against Tenascin R (Tnr; red), β4 spectrin (green), and β3 Tubulin (blue). $N = 3$ independent experiments. Scale bar, 25 μm. **h** Cultured hippocampal neurons transduced at DIV0 with AAV to express Cas9 and control, Nfasc, or Cntn1 gRNAs. Neurons were labeled at DIV21 using antibodies against Tnr (red), β4 spectrin (green), and Nfasc (blue). Scale bar, 10 μm. **i** Quantification of the percentage of transduced neurons with AIS Tnr. $N = 3$ independent experiments. Ordinary one-way ANOVA with Tukey's multiple comparisons test. Error bars, ±SEM. The total number of neurons analyzed is also indicated.

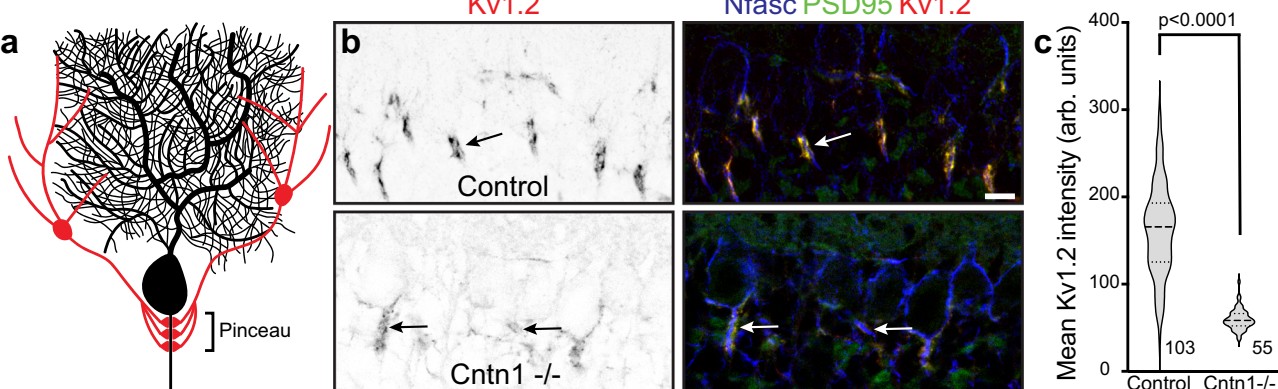

**Fig. 8 | Cntn1 is required for axo-axonic innervation of Purkinje neuron AIS. a** Illustration of a Purkinje neuron (black) with a basket cell (red) forming the cerebellar pinceau on the AIS. **b** Immunostaining of P17 cerebellar pinceau in control and *Cntn1* −/− mouse brain using antibodies against Kv1.2 (red) and PSD95 (green) to label the pinceau, and Nfasc (blue) to label the Purkinje neuron AIS. Scale bar, 10 μm. $N = 3$ control and 2 *Cntn1* −/− mice. **c** Violin plot of the mean Kv1.2

intensity of the cerebellar pinceau in control and *Cntn1* −/− mice. $N = 3$ control and 2 *Cntn1* −/− mice. The number of pinceau analyzed is indicated. In the violin plot the dashed lines indicate the median, while the dotted lines represent the first and third quartiles. $P$ values were calculated using a nonparametric two-sided $T$-test; no adjustments were made for multiple comparisons; ($t = 14.41$, df = 156) $p = 1.8 \times 10^{-30}$.

had not been previously described as AIS proteins. In particular, we found that Cntn1 is highly enriched at the AIS. We validated its enrichment there by endogenous gene tagging in vitro and in vivo, by expression of exogenous epitope-tagged Cntn1 at the AIS in vitro and in vivo, and by immunostaining of cultured hippocampal neurons using Cntn1 antibodies whose specificity was confirmed using CRISPR-mediated gene disruption. The majority of cells we analyzed are excitatory pyramidal neurons, but we cannot rule out the possibility that AIS Cntn1 may be expressed in a cell-type specific manner. This possibility and other candidate AIS proteins (e.g., Tenm4 and Ptprs) will require additional studies to further validate and confirm they are bona fide AIS proteins. Additional endogenous gene tagging may reveal more AIS membrane proteins since we only tested a small subset of the Nfasc proximity proteome.

The use of numerous methods to confirm that Cntn1 is a bona fide AIS protein is important since relying on antibody staining alone can lead to incorrect assignment of a protein being enriched at the AIS[19–21]. Methods allowing CRISPR-mediated endogenous gene tagging are a significant advance to validate protein localization without the confound of off-target antibodies or mislocalization due to over-expression[22–24]. However, the method of tag insertion we used also disrupts coding regions in the last exon of the proteins analyzed, and some proteins that depend on their C-terminal amino acids may be mislocalized. Thus, failure of a protein to localize to the AIS after endogenous gene tagging should not be considered a definitive criterion for exclusion as an AIS protein.

Cntn1 is a GPI-anchored cell adhesion molecule that has been studied in the nervous system mainly in the context of its role in axon-glia interactions as an essential component of the paranodal axoglial junction formed between axons and myelinating glia[31]. There, Cntn1 participates in cis interactions in the axon with Caspr1 (Contactin ASsociated PRotein 1) and trans interactions with the glial 155 kD form of Nfasc (NF155)[30,47]. Cntn1 can engage in diverse interactions with many cell adhesion and extracellular matrix molecules including Caspr1, members of the L1 family of cell adhesion molecules (Nfasc, NrCAM and L1CAM), Tnr, Tnc, and receptor tyrosine phosphatase β[29,48,49]. Our experiments, consistent with other reports[36], show that Cntn1's AIS localization requires its first four Ig-like domains and redundant interactions with either Nfasc or NrCAM (Fig. 10a), since only simultaneous deletion of both disrupts Cntn1's AIS localization. In addition, Cntn1 helps assemble the AIS extracellular matrix since its loss affects Tnr and Bcan recruitment. Future experiments may also reveal roles for Cntn1 in association with other AIS extracellular matrix molecules or membrane proteins including Tnc and Na$^+$ channel β subunits[50]. The importance of Cntn1 in humans is highlighted by the observation that a pathogenic variant of *CNTN1* caused lethal severe fetal akinesia syndrome[51].

ChCs and basket cells precisely innervate cortical PyNs and Purkinje neurons, respectively, to regulate AIS excitability. For example, Dudok et al.[40]. and Schneider-Mizell et al.[41]. showed that a variety of behaviors including pupil dilation, locomotion, and whisking can synchronously activate populations of ChCs to inhibit

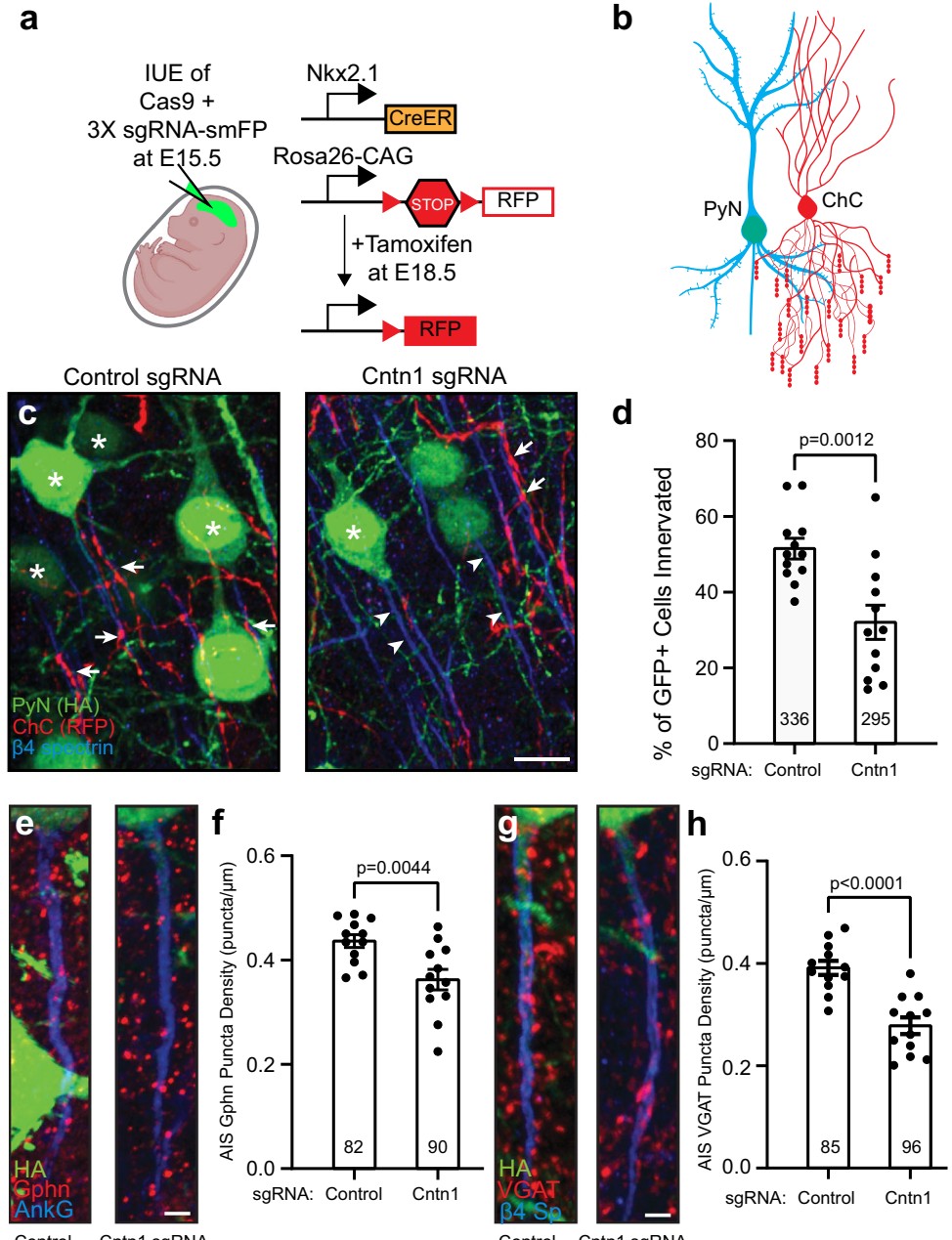

**Fig. 9 | Pyramidal neuron Cntn1 regulates AIS synaptic innervation by ChCs.**
**a** The knockout and labeling strategy for PyN and ChCs. PyNs are electroporated at E15.5 using plasmids to express Cas9 and 3X sgRNA-smFP (HA tag) to delete expression of Cntn1. ChCs are labeled by expression of red fluorescent protein (RFP) using inducible Cre (CreER) in *Nkx2.1-CreER* mice at E18.5. **b** ChC (red) innervation of PyN (blue/green) AIS. **c** PyNs (HA, green) innervated at their AIS (β4 spectrin, blue) by ChC cartridges (red) in layer II of the somatosensory cortex from *Nkx2.1-CreER;Ai9 mice* co-electroporated at E15.5 with a plasmid expressing Cas9 and a plasmid expressing smFP-HA and a control sgRNA or Cntn1 sgRNA; mice were sacrificed at P17. Stars indicate HA+ PyNs and arrows indicate ChC innervation of PyN AISs; arrowheads indicate AIS of PyNs transfected with Cntn1 sgRNA lacking innervation by ChC cartridges. Scale bar, 10 μm. **d** The percentage of HA+ PyNs innervated by single RFP+ ChCs at P17. 12 ChCs and 336 and 295 HA+ PyNs measured from 3 animals were analyzed for Control and Cntn1 sgRNA, respectively.

Data are mean ± SEM. (*t* = 3.701, df = 22) *P* = 0.0012. **e, f** HA+ PyN AISs from *Nkx2.1-CreER;Ai9 mice* electroporated at E15.5 and sacrificed at P17. Inhibitory synapses are visualized by immunostaining for the GABAergic postsynaptic marker gephyrin (Gphn; red; **e**) or the GABAergic presynaptic marker VGAT (red; **g**). AISs (blue) immunostained for AnkG in (**e**) and β4 spectrin in (**g**). Scale bars, 2 μm. **f, h** The average number of gephyrin (**f**) or VGAT (**h**) puncta per μm of HA+ PyN AIS at P17. 82 and 90 HA+ PyN AIS from 3 animals were analyzed for Gphn (**f**) in Control and Cntn1 sgRNA, respectively. 85 and 96 HA+ PyN AIS from 3 animals were analyzed for VGAT puncta (**h**) in Control and Cntn1 sgRNA, respectively. In **f** and **h**, data are mean ± SEM. For **f**, (*t* = 3.176, df = 22) *p* = 0.0044; for (**h**), (*t* = 5.313, df = 22) *p* = $2.0 \times 10^{-5}$. In **d**, **f**, and **h**, *P* values were calculated using a nonparametric two-sided *T*-test; no adjustments were made for multiple comparisons. **a** and **b** Created with Biorender.com.

PyNs through GABAergic synapses. Together, these observations highlight the central role played by ChCs in modulating brain states and behavior. Similarly, pinceau synapses provide strong inhibitory control over Purkinje neuron output, but this is due to ephaptic inhibition rather than chemical inhibition[52]. Despite their importance, the molecular mechanisms responsible for the precise innervation and maintenance of AIS axo-axonic synapses are incompletely understood.

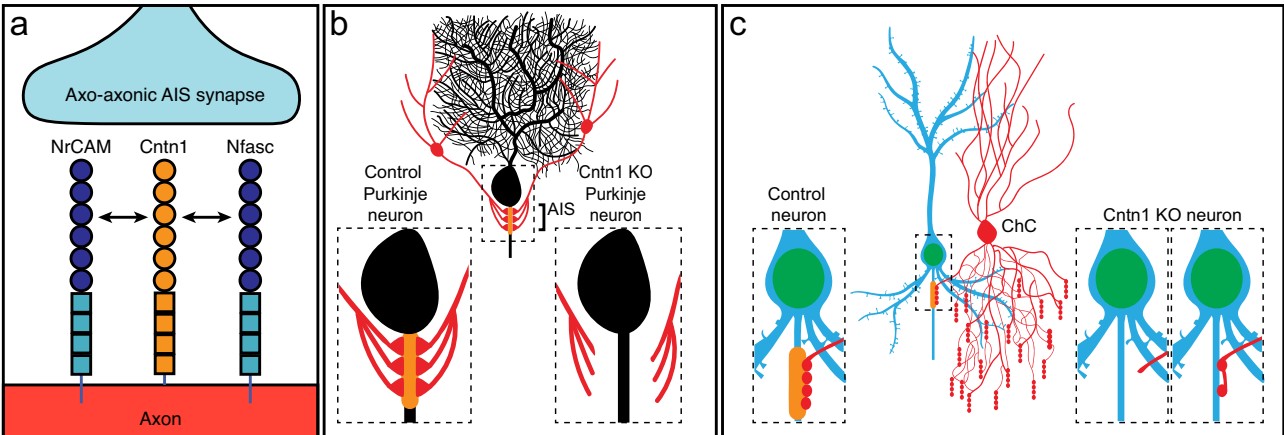

**Fig. 10 | Summary of results. a** Cntn1 interacts with and is redundantly recruited to the AIS through interactions with both NrCAM and Nfasc. **b** Loss of AIS Cntn1 (orange at AIS) from cerebellar Purkinje neurons disrupts basket cell innervation of the AIS and formation of pinceau synapses. **c** Loss of AIS Cntn1 (orange) from Pyramidal neurons results in reduced innervation of AIS by Chandelier cells (ChC) and reduced numbers of AIS inhibitory synapses. Created with Biorender.com.

Loss of AnkG from Purkinje neurons disrupts pinceau synapse formation and has been attributed to mislocalization of AIS NF186; however, loss of ankyrin-binding NrCAM, L1CAM, or CHL1 did not affect pinceau synapse assembly[39]. These observations are consistent with our findings that Cntn1 also regulates pinceau synapse assembly (Fig. 10b), since loss of AnkG affects both NF186 and NrCAM localization[53]. We report here that Cntn1's AIS localization can be independently directed by both of these CAMs. This result is consistent with the observation that the specific deletion of NF186 alone in adults does not disrupt pinceau synapse maintenance[54], suggesting that Cntn1 can also partner with other AIS CAMs (e.g., NrCAM) for synapse maintenance. Since *Cntn1*$^{-/-}$ mice die by P21, we cannot rule out the possibility that pinceau synapses fail to develop due to widespread developmental defects. Future studies utilizing Purkinje neuron specific deletion of Cntn1 will be required to more precisely define Cntn1's role in pinceau synapse assembly and maintenance.

Previously, Tai et al. [43] reported a small RNAi screen of 14 candidate cell adhesion molecules to identify regulators of PyN AIS innervation by ChCs. The candidates screened included Nfasc, NrCAM, and all previously reported AIS CAMs (e.g., Cntn2). Loss of Nfasc, NrCAM, or Cntn2 alone had no effect on ChC/PyN AIS innervation; but the impact of simultaneous loss/depletion of Nfasc and NrCAM remains to be tested. Importantly, among the candidates screened, only loss of L1CAM significantly reduced PyN AIS synaptic innervation by ChCs, despite the fact that L1CAM is found not only at the AIS, but along the entire axon[26]. This suggests that L1CAM may cooperate in cis with other membrane or adhesion molecules like Cntn1 for precise innervation of the AIS and ChC AIS synapse assembly (Fig. 10c). In addition to L1CAM, Hayano et al. [55] reported the cell adhesion molecule Igsf11 functions both pre- and post-synaptically in layer 2/3 of cortex to regulate assembly of PyN/ChC axo-axonic synapses. However, we did not find Igsf11 in our Nfasc-BAR results at any time point; although this may reflect Igsf11 having relatively few extracellular tyrosines. Alternatively, there may be differences between neurons in vitro and in vivo, or even between types of neurons.

One limitation of this study is that the proximity biotinylation experiments were performed on cultured neurons. AIS cell surface proteins whose localization depends on the native brain environment may not be represented. For example, AIS GABAergic synapses may form in vitro, but they are rare, making it difficult to identify presynaptic receptor(s) for Cntn1 and L1CAM using Nfasc-BAR. Future experiments using extracellular split TurboID[56], where PyNs and ChCs each express one half of TurboID may help to identify the pre-synaptic receptor found on ChCs. Alternatively, developing in situ Nfasc-BAR

for use with brain tissue may reveal additional AIS cell surface proteins and their receptors.

In summary, we used extracellular proximity biotinylation to identify Cntn1 as an AIS adhesion molecule. Cntn1 is restricted to the AIS through its binding to AnkG-localized Nfasc or NrCAM (Fig. 10a). In Purkinje neurons, Cntn1 is required for AIS innervation by basket cells (Fig. 10b). In PyNs, Cntn1 functions together with L1CAM to regulate the AIS-specific targeting and developmental assembly of PyN/ChC axo-axonic synapses (Fig. 10c). Thus, our results suggest a model where axo-axonic innervation of diverse neuron types converges on AIS-enriched Cntn1.

## Methods

All experiments were performed in compliance with Baylor College of Medicines Institutional Biosafety Committee. All experiments involving animals were performed in compliance with the National Institutes of Health Guide for the Care and Use of Laboratory Animals and were approved by the Baylor College of Medicine (protocol no. AN4634), the Cold Spring Harbor Laboratories (protocol no. 22-19-16-13-10-06-7), and the Weizmann Institute of Sciences (protocol no. 0053121) Institutional Animal Care and Use Committees.

### Animals

Timed pregnant Sprague-Dawley rats were obtained from Charles River Laboratories. Rats were euthanized for embryo collection at E18. Brains were collected from *Cntn1*$^{-/-}$ and *Cntn1*$^{+/-}$ mice at P17 (catalog #Jax:034216, RRID:IMSR_JAX:034216). The *Cntn1* mice were maintained at the Weizmann Institute of Science, Rehovot, Israel. P0 ICR mice were used for intraventricular injection of AAV for overexpression of Cntn1-Myc. Transgenic Cas9 mice (catalog #Jax: 027650, RRID: IMSR_JAX:027650) were used for intraventricular injection of AAV to perform tagging of endogenous Cntn1. Nkx2-1$^{tm1.1(Cre/ERT2)Zjh}$/J and B6;129S6-Gt(Rosa)26Sor$^{tm9(CAG-tdTomato)Hze}$/J were a gift from Dr. Z.J. Huang[57]. Swiss Webster mice were purchased from Charles River (Cat# CRL:24; RRID: IMSR_CRL:24). There was no consideration of sex for primary neuronal cultures. Cultures were mixed. For *Cntn1*$^{-/-}$ mice sex was also not considered since it was so difficult to obtain *Cntn1*$^{-/-}$ mice due to perinatal lethality, therefore we used whatever sex was available. We also did not consider sex in the in utero electroporation experiments.

### Cell culture

Primary cultures of hippocampal neurons were obtained from E18 Sprague-Dawley rat embryos. Hippocampi were dissected and

dissociated. For imaging, neurons were plated onto Poly-D-Lysine (Sigma) and laminin-coated glass coverslips (Life Technologies) at a density of ~1.25 × 10⁴ cells/cm2. For mass spectrometry, neurons were plated onto Poly-D-Lysine and laminin-coated 10 cm dishes at a density of ~2.5 × 10⁴ cells/cm2. Hippocampal neurons were maintained in Neurobasal medium (Life Technologies) containing 1% Glutamax (Life Technologies), 1% penicillin and streptomycin (Life Technologies), and 2% B27 supplement (Life Technologies) in an incubator at 37 °C with 5% $CO_2$. Half of the media was removed and replaced every 5 days.

## Biotinylation by antibody recognition

Cultured rat primary neurons (~2 × 10⁶ primary hippocampal neurons for each condition) were biotinylated at each of 5 different timepoints. Cells were live labeled by incubating with primary antibodies diluted in culture media for 1 h at 37 °C, then washed with neurobasal media and incubated in culture media alone for 1 h at 37 °C. Cells were incubated with horseradish peroxidase (HRP)-labeled secondary antibodies diluted in culture media for 30 min at 37 °C, then washed with PBS. Biotin tyramide (Perkin Elmer Cat# NEL749A001KT) was diluted 1:500 in a dilution buffer containing $H_2O_2$, and applied to cells for 5 min at 4 °C. Cells were washed with PBS and lysed in RIPA buffer (50 mM Tris-HCl, 150 mM NaCl, 0.5% sodium deoxycholate, 0.1% SDS, and 1% NP-40). Biotinylated proteins were isolated using streptavidin magnetic sepharose beads (GE Healthcare Cat# 28-9857-38) overnight at 4 °C and then washed seven times in RIPA buffer. Control cells were labeled and processed using the same steps except for the omission of primary antibodies.

## Mass spectrometry

Sample-incubated streptavidin sepharose beads were resuspended in 5 mM DTT in 100 mM $NH_4HCO_3$ and incubated for 30 min at room temperature. After this, iodoacetamide was added to a final concentration of 7.5 mM and samples incubated for 30 additional minutes. In all, 0.5 µg of sequencing grade trypsin (Promega) was added to each sample and incubated at 37 °C overnight. Supernatants of the beads were recovered, and beads digested again using 0.5 µg trypsin in 100 mM $NH_4HCO_3$ for 2 h. Peptides from both consecutive digestions were recovered by solid phase extraction using C18 ZipTips (Millipore, Cat# ZTC18S096), and resuspended in 0.1% formic acid for analysis by liquid chromatography-mass spectrometry (LC-MS/MS). Peptides resulting from trypsinization were analyzed in a Orbitrap Lumos Fusion (Thermo Scientific), connected to a NanoAcquity™ Ultra Performance UPLC system (Waters). A 15-cm EasySpray C18 column (Thermo Scientific) was used to resolve peptides (90 min gradient with 0.1% formic acid in water as mobile phase A and 0.1% formic acid in acetonitrile as mobile phase B). MS was operated in data-dependent mode to automatically switch between MS and MS/MS. MS spectra were acquired between 375 and 1500 m/z with a resolution of 120000. For each MS spectrum, multiply charged ions ions over the selected threshold (2E4) were selected for MS/MS in cycles of 3 s with an isolation window of 0.7 m/z. Precursor ions were fragmented by HCD. MS/MS spectra were acquired in centroid mode with resolution 60000 from m/z = 110. A dynamic exclusion window was applied which prevented the same m/z (mass tolerance 30 ppm) from being selected for 30 s after its acquisition. Peak lists were generated using PAVA software[58]. All generated peak lists were searched against the mouse subset of the SwissProt database (SwissProt.2017.11.01) using Protein Prospector[59]. The database search was performed with the following parameters: a mass tolerance of 10 ppm for precursor masses; 30 ppm for MS/MS, cysteine carbamidomethylation as a fixed modification and acetylation of the N terminus of the protein, pyroglutamate formation from N-terminal glutamine, and oxidation of methionine as variable modifications. All spectra identified as matches to peptides of a given protein were reported, and the number of spectra (Peptide Spectral Matches,

PSMs) used for label free quantitation of protein abundance in the samples.

## Antibodies

The following antibodies were used for in vitro biotinylation (dilutions for each antibody are indicated in parentheses): chicken polyclonal antibody anti-neurofascin (1:1000; R&D Systems Cat# AF3235, RRID:AB_10890736), rabbit polyclonal anti-NrCAM (1:1000; Abcam Cat# ab24344, RRID:AB_448024), anti-chicken HRP-labeled secondary antibody (1:2000; Aves Labs Cat# H-1004, RRID:AB_2313517), anti-rabbit HRP-secondary antibody (1:2000; Jackson ImmunoResearch Labs Cat# 111-035-003, RRID:AB_2313567).

The following antibodies were used for immunofluorescence studies (dilutions for each antibody are indicated in parentheses): chicken polyclonal anti-neurofascin (1:500; R&D Systems Cat# AF3235, RRID:AB_10890736), chicken polyclonal anti-MAP2 (1:1000; EnCor Biotechnology Cat# CPCA-MAP2, RRID:AB_2138173), mouse monoclonal anti-Ankyrin-G (1:500; NeuroMab N106/36, RRID:AB_10673030), mouse monoclonal anti-Tenascin-R (1:250; R&D Systems Cat# MAB1624, RRID:AB_2207001), rabbit polyclonal anti-NrCAM (1:250; Abcam Cat# ab24344, RRID:AB_448024), rabbit polyclonal anti-β4 spectrin (1:500, Rasband lab, RRID:AB_2315634), rabbit polyclonal anti-Kv1.2 (1:250, James Trimmer, University of California, Davis, RRID:AB_2756300), mouse monoclonal anti-PSD-95 (1:250; Antibodies Incorporated Cat# 75-028, RRID:AB_2292909), mouse monoclonal anti-Tuj1 (1:700; BioLegend Cat# 801202, RRID:AB_10063408), goat polyclonal anti-Cntn1 (1:500; R&D Systems Cat# AF904, RRID:AB_2292070), mouse monoclonal anti-Myc (1:2000; MBL International Corporation Cat# M192, PRID: AB_11160947), rat monoclonal anti-HA (1:500; Millipore Sigma Cat# 11867423001, RRID: AB_390918), mouse monoclonal anti-V5 (1:500; Invitrogen Cat# R960CUS, RRID: AB_159298). Anti-RFP (guinea pig pAb, 1:1000, Synaptic systems 390 005), anti-gephyrin (mouse mAb IgG1, 1:500, Synaptic Systems 147 011), anti-VGAT (guinea pig pAb, 1:500, Synaptic Systems 131 004), and anti-Brevican (guinea pig pAb, 1:250, Constanze Seidenbecher, University Magdeburg).

The following secondary antibodies were used: Alexa Fluor 555 goat anti-rat (1:1000, Thermo Fisher Scientific A-11006), Alexa Fluor Plus 555 goat anti-rabbit (1:1000, Thermo Fisher Scientific A32732), Alexa Fluor 647 goat anti-rabbit (1:1000, Thermo Fisher Scientific A-21244), Alexa Fluor 555 goat anti-guinea pig (1:1000, Thermo Fisher Scientific A-21435), Alexa Fluor 647 goat anti-guinea pig (1:1000, Thermo Fisher Scientific A-21450), Alexa Fluor 647 goat anti-mouse IgG1 (1:1000, Thermo Fisher Scientific A-21240), Alexa Fluor 488 goat anti-mouse IgG2a (1:1000, Thermo Fisher Scientific A-21131), Amino-methylcoumarin (AMCA) anti-chicken IgY (1:1000 Jackson Immunoresearch labs 103-155-155), Alexa Fluor 488 anti-chicken IgY, (1:1000 Jackson Immunoresearch labs 103-545-155), Alexa Fluor 488 anti-mouse IgG (1:1000 Thermo Fisher Scientific A11029), Alexa Fluor 594 anti-rabbit (1:1000 Thermo Fisher Scientific A11034), Alexa Fluor 594 anti-mouse IgG (1:1000 Thermo Fisher Scientific A32742). Streptavidin Alexa Fluor 594 conjugates were purchased from Thermo Fisher Scientific (1:5000; S11227). Hoechst fluorescent reagent (1:100,000; Thermo Fisher Scientific Cat# H3569, RRID:AB_2651133) was used to label nuclei.

## In utero electroporation and tamoxifen induction

To manipulate Cntn1 expression in pyramidal neurons (PyNs) and sparsely label chandelier cells (ChCs) in the same neocortical layer, ventricular zone-directed in utero electroporation targeting neocortical PyN progenitors was performed in *Nkx2.1-CreER;Rosa26-loxp-STOP-loxp-tdTomato (Ai9)* embryos. Specifically, Swiss Webster females were bred with *Nkx2.1-CreER⁺/⁻;Ai9⁺/⁺* males. Pregnant females at 15.5 days of gestation were anesthetized, the uterine horns were exposed, and ~1 µl of plasmid solution (0.75 µg/µl pCAG-1BPNLS-Cas9-

1BPNLS + 1.5 μg/μl pAAV-3x-sgRNA-smFP (with control or target specific sgRNAs; see STAR Methods) was injected manually into the lateral ventricle of the embryos using a beveled glass micropipette (Drummond Scientific). After injection, five square 50 ms pulses of 45 V with 950 ms intervals were delivered across the uterus with two 5 mm electrode paddles (BTX, 45-0489) positioned on either side of the head (BTX, ECM830). After electroporation, the uterine horns were placed back in the abdominal cavity of the pregnant dam and the wound was surgically sutured. Tamoxifen (3 mg/30 g of body weight) was administered to the pregnant dam by oral gavage at 18.5 days of gestation to induce CreER activity and excision of the STOP cassette, resulting in tdTomato red fluorescent protein expression in a sparse population of nascent neocortical ChCs in their offspring. Pups were euthanized at postnatal day 17.

### Immunofluorescence labeling

Cultured rat primary hippocampal neurons were fixed in 4% paraformaldehyde (PFA, pH 7.2) for 15 min at 4 °C. Acutely dissected brains were drop fixed in 4% paraformaldehyde (PFA, pH 7.2) for 60 min at 4 °C. Brains were then equilibrated in 20% and 30% sucrose in 0.1 M Phosphate Buffer (PB) overnight at 4 °C. Brains were then sectioned at 12–25 μm and mounted on coverslips. Fixed neurons and brain sections were permeabilized and blocked with 10% normal goat serum in 0.1 M PB with 0.3% Triton X-100 (PBTGS) for 1 h. In the case where we used human anti-Cntn1 antibodies, blocking was performed with Gelatin (PBTGel [0.22% gelatin and 0.3% Triton X-100 in 0.1M PB]). Cells and sections were then incubated in primary antibodies diluted in PBTGS overnight at room temperature or 4 °C. Tissues and cells were then washed three times using PBTGS for 5 min. each. Fluorescent secondary antibodies were then diluted in PBTGS and added to cells and tissues for 1 h. Coverslips were then washed once using PBTGS, 0.1 M PB, and finally 0.05 M PB for 5 min. each. Coverslips were then mounted using Vectashield plus (Vector Labs) anti-fade mounting media.

For immunostaining of electroporated neonatal brains, P17 mice were deeply anesthetized with isoflurane and perfused transcardially with PBS and 4% paraformaldehyde (PFA) in 0.1 M phosphate buffer. Brains were post-fixed in 4% PFA in 0.1 M phosphate buffer overnight at 4 °C and then cryoprotected with 30% sucrose in PBS. 50 μm thick coronal sections were subsequently generated using a Vibratome (Leica VT1000S). For gephyrin and VGAT immunostaining, brain slices were subjected to mild antigen retrieval with 10 mM citrate buffer for 30 min at 60 °C. Subsequently, brain slices were blocked and permeabilized with 10% normal goat serum (NGS) and 0.3% Triton X-100 in PBS at RT for 30 min and then incubated with primary antibodies diluted in 2% NGS and 0.3% Triton X-100 in PBS overnight at 4 °C. Fluorescent secondary antibodies diluted in 2% NGS and 0.3% Triton X-100 in PBS were applied for 2 h at RT the following day. Sections were then washed three times with PBS for 20 min per wash and mounted with Fluoromount-G (Southern Biotech).

### Plasmid construction

The sgRNAs and the homology-independent donor templates were generated following strategies similar to those described[20,23,24,60]. Here, the U6 promoter and scaffold sequences were PCR amplified from pMJ117 and pMJ179 (gifts from Jonathan Weissman, Addgene plasmid #85997 and #85996). The smFP-V5 (a gift from Loren Looger, Addgene plasmid #59758) was used as a knock-in donor.

The knockout constructs expressing three independent sgRNAs and a smFP-HA marker (pAAV-3x-gRNA-smFP) were generated as follows: the U6 promoter and scaffold sequences were PCR amplified from pMJ114, pMJ117, and pMJ179 (gifts from Jonathan Weissman, Addgene plasmid # 85995, #85997, and #85996). Human Synapsin1 promoter and smFP-HA were PCR amplified from pAAV-hSyn-EGFP (a gift from Bryan Roth, Addgene Plasmid #50465) and pCAG_smFP-HA (a

gift from Loren Looger, Addgene plasmid #59759), respectively. The plasmid PX552 (a gift from Feng Zhang, Addgene plasmid #60958) was digested with a NotI restriction enzyme (NEB) and used as a plasmid backbone. DNA fragments were ligated together using an In-Fusion Snap Assembly Master Mix (Takara). The sgRNA sequences for knock-in and knockout are listed in supplementary data 4. The AAV-SpCas9 plasmid (a gift from Feng Zhang, Addgene plasmid #60957) was modified by removing the HA tag.

Cntn1 constructs were generated in both pcDNA3 and AAV backbones. pcDNA3 was digested with EcoRI restriction enzyme (NEB) and pAAV-hSyn-EGFP (a gift from Bryan Roth, Addgene Plasmid #50465) was digested with BamHI and XhoI restriction enzymes (NEB) and used as plasmid backbones. Full-length and truncated Cntn1 was PCR amplified from rat contactin-myc[35] and ligated together using an In-Fusion Snap Assembly Master Mix (Takara). All DNA constructs were verified by sequencing (Genewiz and plasmidsaurus).

### Adeno-associated virus (AAV) production

Small scale AAV cell-lysates were produced using the AAVpro Purification Kit (All Serotypes) (Takara) with slight modifications. Briefly, HEK293T cells were triple-transfected with AAV plasmid, helper plasmid (Agilent Technologies, Cat # 240071), and serotype PHP.S or PHP.eB plasmids (a gift from Viviana Gradinaru, Addgene plasmids #103002 and #103005) with PEI Max (Polysciences, Cat # 24765). The medium was changed the next day of transfection and cells were incubated for 3 days after transfection. HEK cells were then collected and lysed with the AAV Extraction Solution A plus. The extracted solution was centrifuged at 10,000 x g for 10 min to remove debris and mixed with Extraction Solution B. This small scale AAV solution was stored at 4 °C and used for neuronal transduction into cultured neurons. AAV vectors for in vivo transduction were produced by the Baylor College of Medicine Neuroconnectivity Core or in our lab following the strategies described previously[61].

### Viral transduction of neurons

For viral transduction of cultured neurons, 10 μl of AAV-Cas9 and 10 μl of AAV-sgRNA and donor, or AAV-3x-sgRNA-smFP, were added into a well of a 12-well plate at 0–1 DIV. The medium was replaced 2 days after infection. For viral transduction of neurons in vivo, AAV vectors were injected into the lateral ventricles of neonatal mice as described previously[62]. Briefly, P0 to P2 pups were anesthetized on ice and 1–2 μl of AAV vectors were bilaterally injected. The pups were placed in a heated cage until the animals recovered and then returned to their mother. For transduction of neurons in adult, viruses were injected retro-orbitally in 13-week-old C57Bl/6 J mice. Tissues were collected 4 weeks after infection.

### Confocal image acquisition and analysis of ChC/PyN AIS innervation and GABAergic synapse density at PyN AISs

For analysis of ChC/PyN AIS percent innervation, images of coronal brain slices (50 μm thick) were acquired using an LSM 800 confocal laser-scanning microscope (Zeiss) with a 63× oil-immersion objective and sequential acquisition settings applied at a resolution of 1024 × 1024 pixels. 200 μm × 200 μm images of single RFP+ ChCs and neighboring GFP+ electroporated PyNs in layer II of the somatosensory cortex were collected using a z series of 30–36 images with a depth interval of 1 μm. ChC/PyN AIS percent innervation was calculated by dividing the number of GFP+ PyNs innervated at their AIS by a single RFP+ ChC by the total number of GFP+ PyNs in the entire 200 μm × 200 μm image z stack. A PyN AIS is considered innervated when there are at least two ChC boutons from the same cartridge present at the AIS. Representative maximum projection images were generated from 10 z-series images with a depth interval of 1 μm. To quantify the average density of gephyrin or VGAT puncta per μm on the AIS of PyNs, 90 μm × 90 μm images were acquired at a resolution of

1280 × 1280 pixels using a z-series of 40–60 images with a depth interval of 0.37 μm. The number of gephyrin or VGAT puncta overlapping with AnkG+ or β4-spectrin+ PyN AISs in individual z-plane images was manually counted and AIS lengths were measured using Zeiss Zen (Blue Edition) imaging software. A punctum was considered positive if the area of the immuno-positive signal for gephyrin or VGAT fell within the range of ~0.1–0.5 μm² and at least half of it overlapped with the AnkG or β4 spectrin immunosignal that outlines the AIS. In addition, the fluorescence intensity of the punctum had to be more than two-fold higher than the surrounding background. Gephyrin or VGAT puncta density at the AIS was then calculated by dividing the number of PyN AIS gephyrin or VGAT puncta by the length of the AIS. Representative maximum projection images of PyN AIS GABAergic synapses visualized via gephyrin or VGAT immunostaining were generated using a z-series of 10 images with a depth interval of 0.37 μm.

The distance from the soma to the start of the AIS was determined by measuring the distance from the start axon hillock to the onset of β4 spectrin immunoreactivity. For quantification of endogenous β4 spectrin expression levels at HA+ PyN AISs, integrated fluorescence intensity of AIS-localized β4 spectrin was calculated using ImageJ software. The AIS (region of interest (ROI)) was outlined and the mean fluorescent intensity (mFI) of β4 spectrin within the ROI was measured. In addition, the mean fluorescence intensity of the background outside the ROI was measured for that channel. Integrated fluorescence intensity was calculated by using the following formula: (ROI mFI – background mFI) × ROI area. All image analyses were performed by an observer blinded to the experimental conditions.

### Image acquisition and analysis
Images of immunofluorescence were captured using an Axio-imager Z2 microscope fitted with an apotome attachment for structured illumination (Carl Zeiss MicroImaging) and a Nikon Eclipse Ni2. 20X (0.8 NA), 40X (0.95 NA), and 63X (1.4 NA) objectives were used. Images were taken using Zen 3.2 (Zeiss) or NIS-Elements (Nikon). For measurements of AIS streptavidin fluorescence intensity, 20 neurons per timepoint per replicate were imaged and line scans were drawn using Zen 3.2 software. AIS were identified by immunostaining using known AIS proteins (AnkG, β4 spectrin, Nfasc, or NrCAM). For the analysis of cerebellar pinceau, a region of interest including the Purkinje neuron AIS (labeled by Nfasc) was manually drawn. The fluorescence intensity for Kv1.2 was measured for each region of interest and normalized to the area. All measurements were taken with the same exposure times and immunolabeling was also performed at the same time. Images were exported to Fiji, Adobe Photoshop, and Adobe illustrator for figure presentation. Some figures were generated using Biorender.

### Statistics and reproducibility
Unpaired, two-tailed Student's $t$ test was used for all statistical analyses unless otherwise indicated. Data were analyzed using Microsoft Excel and GraphPad Prism. All error bars are ±SEM unless otherwise indicated. PSMs were normalized using the formula $PSM_{Norm} = PSM_{DIVX} * Csum_{Max}/Csum_{DIVX}$. $PSM_{DIVX}$ is the raw PSM count for a candidate protein for a particular replicate at the specified timepoint; $Csum_{Max}$ is the maximum sum of seven endogenously biotinylated carboxylases Acaca, Acacb, Pc, Pcca, Pccb, Mccc1, Mccc2 for an individual replicate across all 15 biological replicates; $Csum_{DIVX}$ is the sum of seven endogenously biotinylated carboxylases Acaca, Acacb, Pc, Pcca, Pccb, Mccc1, Mccc2 for an individual replicate at the specified timepoint DIVX. Heatmaps were generated using GraphPad Prism. Candidates were rank-ordered by the slope of the linear regression of their $log_2$ fold enrichment over time. The number of extracellular tyrosines for proteins whose $log_2$ fold change was >2 and that had at least ten PSMs were counted using a script that extracted protein topology from the Uniprot database [www.uniprot.org]. The tyrosine-PSM proximity plot was generated using Python [www.python.org]. Gene Ontology analyses were performed using ShinyGO

v0.77 [http://bioinformatics.sdstate.edu/go/]. Colors were added in Adobe Illustrator.

### Extended materials
A detailed list of all materials including all gRNA sequences, antibodies, plasmids, sources, etc. is provided in the Supplementary Information file.

### Reporting summary
Further information on research design is available in the Nature Portfolio Reporting Summary linked to this article.

## Data availability
All data generated or analyzed during this study are included in this published paper. All proteomics data are available via ProteomeXchange with identifier PXD043805. Supplementary material including all PSMs are including in the Supplementary Data and Supplementary Information. The source data underlying Figs. 4c–d, 6d, h, k, 7d, f, i, 8c, 9d, f, h, S1b, S6a–d, S7c–d, S8a–c are provided in the Source Data file. Source data are provided with this paper.

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

## Acknowledgements

The work reported here was supported by NIH research grants NS122073 (M.N.R.), MH119819 (L.V.A.) and NS116897 (L.V.A.), and by the Dr. Miriam and Sheldon G. Adelson Medical Research Foundation (A.L.B., E.P., and M.N.R.). We thank Ayano Ogawa for help with illustrations.

## Author contributions

Conceptualization methodology, validation, investigation, visualization, and writing—reviewing and editing: Y.O. and B.C.L.; Investigation, analysis, and writing—reviewing and editing: S.G.; Investigation and analysis: J.M.R., Y.E-E., H.H., S.N., and F.B.; Investigation, data curation: J.O-P.; Resources, supervision, funding acquisition, and writing—reviewing and editing: A.L.B., E.P., and L.V.A.; Conceptualization, methodology, data curation, writing—original draft and editing, project administration, and funding acquisition: M.N.R.

## Competing interests

The authors declare no competing interests.
