## [Peer Review File · Nature Communications]

Antibody-directed extracellular proximity biotinylation reveals Contactin-1 regulates axo-axonic innervation of axon initial segmentsREVIEWER COMMENTS

Reviewer #1 (Remarks to the Author):

This is an extremely interesting manuscript which first describes a proximity biotinylation-based screen for identifying novel proteins localised near the extracellular domain of neurofascin-186, then goes on to establish that one of these candidates – Contactin-1 (Cntn1) – is truly localised and concentrated at the AIS. Moreover, state-of-the-art experimental manipulations show that Cntn1 contributes to the formation of the AIS extracellular matrix, and to the development of axo-axonic synaptic inputs in two distinct cell types in the intact brain. These are all novel and important findings. The data are largely convincing and the manuscript is very clearly presented. I have just a few issues that I feel should be addressed in order for the authors to be able to justify all of the claims they make.

Major

- Lack of quantification. Figure 6e/h is the first instance of any quantitative analysis in the paper. Prior to this, a great many results are presented as statements backed only by a single example image. Ideally the authors would use a quantitative, objective metric of 'AIS localisation' which could then be applied across multiple neurons in all conditions in an unbiased manner. At the very least, the %-based quantification presented in Fig6e/h should be applied to all previous reported observations in Fig4c-k, Fig5, and Fig6a-c. This analysis should be done by experimenters blind to experimental condition, and should be accompanied by an explicit, detailed description in the Methods as to how a 'neuron with AIS label' is defined.

- Effects of manipulations on the AIS itself. When Cntn1 levels are reduced in cultures (Fig 5a) or in vivo (Figs7, 8), does the AIS change in terms of its length, position, and/or labelling intensity for e.g. AnkG/NFasc/b4spectrin? The AIS labelling has already been carried out in each case, and in Fig 8 the AIS length data has already been acquired to calculate synaptic input densities, so this should not require a great deal of extra work. It is, however, important for the interpretation of Cntn1's role at the AIS – can some of the effects of Cntn1 depletion be due to indirect effects on the structure of the AIS itself?

- Effectiveness of the Cntn1 manipulations. The depletion of Cntn1 with CRISPR-based methods is never quantified. This can be readily done with antibody labelling in culture (e.g. quantify/count Cntn1+ AISs). In ex vivo tissue the lack of good Cntn1 antibody labelling makes this difficult, but it is important nevertheless to know that the manipulated cells did in fact have much reduced levels of Cntn1 (i.e. experimental results are not due to non-specific features of the constructs). Perhaps RNAscope or similar mRNA quantification approaches might be able to demonstrate this?

Minor

- In the Abstract (line 39) the line saying that Cntn1 is 'required for' axo-axonic innervation should be toned down, given that there is still plenty of innervation after Cntn1 manipulation.

- Might the presence of Cntn1 at the AIS be cell-type-specific? Fig6e shows ~70% of cells have (myc-tagged OE) Cntn1 at the AIS in culture, and most example images look like large, presumptive pyramidal neurons. Are the other 30% all GABAergic, for example? Or Prox1+ DGCs? It would be good to at least discuss these possibilities in the text.

- The example images also feature many double-axon cells (e.g. Fig4e,f,k; Fig5b). These are certainly common in culture, but is this purely coincidence, or is it a feature of cells that have strong Cntn1 at the AIS?

- Methods: was the culture medium ever changed?

- Methods: how were AISs imaged and measured?

- Methods: how was the Kv1.2 intensity analysis carried out? How was tissue from both conditions treated to minimise variability in tissue quality/fixation, staining and image acquisition? How were pinceau identified for analysis, and how was fluorescence intensity analysed? The number of quantified synapses in the mutant group is nearly half of those of the controls. Is this because there was an overall loss of these synapses besides a disruption? Why are there only 2 quantified mutants as compared to the 3 controls? Is this because they are sick?
- Methods for Fig 8d: how was an innervated AIS defined objectively?
- Methods for Fig 8f/h: what were the objective criteria for inclusion as a punctum? And as 'co-localised' with AIS label? Was this, and all other analysis, carried out blind?
- Figure S1a: please annotate the images of the panel. What is the signal detected here? Is it streptavidin?
- Lines 135 and 136: The sentence 'all experiments were performed three times independently for each developmental timepoint with 2 million neurons per experiment' should be moved to the corresponding figure legend or methods.
- Figure S2: the authors should specify what 'replicates' mean in this context: are these technical replicates (e.g., plates) or biological replicates (cultures)?
- Lines 176 and 177: The statement 'Similarly, Nav1.2 (encoded by Scn2a) the main voltage-gated Na⁺ channel expressed at the AIS in DIV12 hippocampal neurons' should include a reference to existing literature.
- Figure 5. It is unclear at which stage of development (DIV) the neurons were infected, as well as when the immunostaining was performed. Please specify in the corresponding part of the text/figure legends.
- Line 254 and figure 5: the authors refer to the labelling showed in figure 5f as 'cortical neurons.' Please specify which area and layer of cortex you are showing.
- Figure legend on Figure 5: explanation for the 'RO' and Tg abbreviations is missing in the figure legend.
- Figure 6: the neuronal stage/DIV of the pictures showing the immunostaining is not stated either in the results nor in the figure legend. This applies for the stage at which the infection was performed.
- Line 289: to match the previous writing format, please change 'loss of both Nfasc and Cntn1' to 'loss of Nfasc or Cntn1'.
- Figure 7c: please indicate in the figure legend what do the different lines in the violin plots indicate (e.g., mean).
- Line 317: replace 'remains' by 'remain'.
- Figure S5c: the results of the paper show that the deletion of Cntn1 leads to 1) a decrease in the percentage of pyramidal neurons innervated by chandelier cells (figure 8d) and 2) a reduction of inhibitory synapses between the two cell types (figure 8f and 8h). However, in Figure S5c, it seems that only the effect 1) is showed. The schematic would be more accurate if the second scenario is also pictured.
- Line 582: please specify the 'PB' abbreviation.
- Line 584: please specify the 'PM' abbreviation.
- Line 589: please capitalise 'Vector Labs' accordingly.

Reviewer #2 (Remarks to the Author):

In this manuscript, Ogawa et al. use antibody based, HRP driven proximity biotinylation to probe the cell surface proximal proteome of the axon initial segment (AIS) protein Neurofascin. Neurofascin proximal proteins emerging from this screen were validated using CRISPR based tagging of endogenous proteins and imaging. The authors then proceeded to further functionally validate one hit emerging from the screen, Cntn1. The authors confirm that Cntn1 is a bona fide AIS protein. Localization of Cntn1 to the AIS requires AnkG-binding L1-family cell adhesion molecules. Finally, the authors show that Cntn1 contributes to assemble of the AIS extracellular matrix and drives AIS axo-axonic innervation by specific brain cells in the cerebellum and the cortex. This is in general a solid

study that is of high quality and of great value to the community. There are, however, a few technical concerns that need to be addressed prior to publication

Main comments:

- The authors make use of a specific antibody targeting the ectodomain of Nfasc to probe its proximal proteome. Since this approach greatly depends on the specificity of the used antibody, the authors also perform a second labeling experiment in which they targeting another AIS protein, NrCAM (Fig S1C and S1D). Comparison of both antibodies reveals a significant overlap (Fig S1D). It would be informative to visualize the obtained data for both antibodies in a Venn diagram which visualizes the overlap with regards to specific hits that were obtained using both antibodies. Did the authors filter out proteins that were identified with just one of the two used antibodies? The intersect of both antibodies represent high confidence AIS proteins. An alternative approach would be to use a different antibody targeting the same protein (Nfasc) or to perform the experiment in cells lacking the primary bait to filter out potential false positive hits.
- The authors should perform GO enrichment analysis of their hits to confirm that these are strongly enriched for extracellular proteins.
- The 'DIV4' experiment had only 63 identified hits (Fig. S3). Is this technical or biological? The authors may want to repeat this experiment. Furthermore, the authors should plot all their obtained data using hierarchical clustering
- The authors state that ion channels are unexpectedly absent in their Nfasc cell surface proximal proteome and state this may be due to the small number of extracellular tyrosine residues in these ion channels, which are the substrates of the biotinylation reaction. Have the authors investigated whether these ion channels are efficiently solubilized in their lysates? They used a fairly stringent lysis buffer but perhaps the ion channels still end up in the insoluble pellet after extraction (did they spin down the lysate after lysis, unclear from the M&M)? This can be easily investigated using western blotting for these ion channels in the input lysate used for streptavidin enrichment.
- The authors validated 23 hits using CRISPR based tagging and immunofluorescence (figure 4). Were these 23 hits the only proteins the authors tagged and validated? Or did they also tag additional proteins which subsequently could not be validated to localize to the AIS?

Reviewer #3 (Remarks to the Author):

In this manuscript, Ogawa et al. focus on the identification of cell surface proteins present at axon initial segments (AIS). Among the candidate AIS proteins they identify through proximity labeling/mass spectrometry, they identify Cntn1 as an AIS protein. Moreover, the authors provide evidence suggesting that Cntn1 promotes axoaxonic innervation at AIS between chandelier cells and pyramidal neurons. Overall, these findings will be significant for two reasons: (i) the authors establish a methodology to identify cell surface proteins within the vicinity of another (in this case neurofascin), (ii) the conclusion that Cntn1 is an AIS protein is novel and helps further define the physiological roles it plays in the nervous system.

With a few exceptions, the work is supported by the experiments reported in this manuscript and there is confidence that the findings reported here will stand the test of time. With that said, there are weaknesses in the approach that the authors should consider addressing before publication, which I support. These are listed below in the order in which they arose in the manuscript.

1. The volcano plots in Fig. 2 are illegible (minor concern)
2. It is not at all certain that the analysis of PSM numbers vs. number of extracellular tyrosine residues is pertinent because a tyrosine could be in the ectodomain and yet not be exposed to solvent. A quick look at the model of Ig1-Ig4 of Cntn1 in AlphaFold revealed that there are quite a few of these

residues.

3. It is probably an overstatement to suggest that Cntn1 is recruited by Nfasc/Nrcam at AIS with the data that is presented (lines 278/9). Although it is true that Nfasc and Nrcam bind to Cntn1 based on recent analyses of interactome data, the authors are not proving that Cntn1 binds to either of those proteins at AIS. One idea would be to repeat the experiments of Fig. 6b using a Cntn1 construct that cannot bind to Nfasc. Residues at the Nfasc/Cntn1 interface that disrupt the interactions between the two proteins can be found in PubMed ID # 36329006.

4. The authors suggest that removal of Cntn1 alters the AIS matrix, but really their evidence points to a decrease in the recruitment of Tnr. This conclusion would be strengthened if the authors analyzed the recruitment of brevican as well.

5. It seems that Fig. S5 could be used as a final figure in the main manuscript (minor concern).

Reviewer #1

Major points:

1. *Lack of quantification.*
 - a. We added quantitation to all experiments used to validate Cntn1 enrichment and knockout at the AIS. These results are included in new Figures 6d, h, and k.
 - b. We respectfully emphasize that all mass spectrometry results are repeated in triplicate with volcano plots generated from fold-enrichment and p-values to test for significance. Our mass-spectrometry results are highly quantitative and the analyses are illustrated in Figure S1b, revised Figure 2, and Figure S3.
 - c. The reviewer suggests adding quantification for Figs. 6a-c. We re-analyzed our truncation analysis and now show the percentage of neurons with AIS Cntn1-myc or the indicated truncation variants. This is now shown in a revised Fig. 7d.
 - d. The reviewer suggests additional quantification for Fig. 4 (now Fig. 5), instead of the subjective evaluation shown in Fig. 5a. Nevertheless, we respectfully disagree with the reviewer that a detailed quantification is valuable – our AAV and CRISPR-mediated tagging strategy was intended to reveal AIS enriched proteins. This was accomplished as illustrated in Figures 5c, d, and k. AIS enrichment or AIS localization (defined as colocalization with β 4 spectrin) is evident by eye. We did not attempt to analyze any other cell surface proteins except those that are clearly enriched – even without taking line scans or ratios between AIS fluorescence and dendritic fluorescence. We could make these kinds of line scan plots, but we do not agree that this would add any value to the manuscript or figures, and would just clutter the figures. In addition, the point of the paper is our discovery of Cntn1 as a new AIS cell surface protein that regulates axo-axonic targeting.
 - e. We used well-known markers of the AIS (e.g. β 4 spectrin as shown in Fig. 5) as our objective comparison to determine if a protein is enriched or located at the AIS. This is stated in the methods.
2. *Do the manipulations have any effect on the AIS itself?*
 - a. We performed sgRNA mediated knock-out of Cntn1 in vitro and immunostained for AnkG, NF186, Na⁺ channels, and β 4 spectrin. We now show there is no change in AIS length, or intensity (new Fig. S5). We also analyzed AIS in vivo after loss of Cntn1. We found no change in AIS length, fluorescence intensity, or position (new Fig. S8).
3. *How efficient is the CRISPR-mediated knockout of Cntn1?*
 - a. We quantified the efficiency of our knockout compared to a control sgRNA. These results are now shown in Fig. 6d.

Minor points:

1. *In the Abstract (line 39) the line saying that Cntn1 is 'required for' axo-axonic innervation should be toned down, given that there is still plenty of innervation after Cntn1 manipulation.*
 - a. Done. We now state "Cntn1 contributes to assembly of the AIS-extracellular matrix, and regulates AIS axo-axonic innervation..."
2. *Might the presence of Cntn1 at the AIS be cell-type-specific? Fig6e shows ~70% of cells have (myc-tagged OE) Cntn1 at the AIS in culture, and most example images look like large, presumptive pyramidal neurons. Are the other 30% all GABAergic, for example? Or Prox1+ DGCs? It would be good to at least discuss these possibilities in the text.*
 - a. This is an interesting idea. And we have added this possibility to the discussion.
3. *The example images also feature many double-axon cells (e.g. Fig4e,f,k; Fig5b). These are certainly common in culture, but is this purely coincidence, or is it a feature of cells that have strong Cntn1 at the AIS?*

- a. As the reviewer correctly notes, neurons with multiple axons are quite common. We see AIS Cntn1 in cells with single axons. It is only a coincidence that we selected a few neurons with multiple axons.
4. *Methods:*
- a. *was the culture medium ever changed?*
- i. Yes, half of the media was removed and replaced every five days. This is now stated in the methods.
- b. *how were AISs imaged and measured?*
- i. All AIS were identified using previously described AIS markers (e.g. $\beta 4$ spectrin, AnkG, Nfasc, and NrCAM). This is now stated in the methods.
- c. *how was the Kv1.2 intensity analysis carried out? How was tissue from both conditions treated to minimise variability in tissue quality/fixation, staining and image acquisition? How were pinceau identified for analysis, and how was fluorescence intensity analysed? The number of quantified synapses in the mutant group is nearly half of those of the controls. Is this because there was an overall loss of these synapses besides a disruption? Why are there only 2 quantified mutants as compared to the 3 controls? Is this because they are sick?*
- i. We apologize for leaving out this description in the methods. The section on Image Acquisition and Analysis now includes: "For the analysis of cerebellar pinceau, a region of interest including the Purkinje neuron AIS (defined by Nfasc labeling) was manually drawn. The fluorescence intensity for Kv1.2 was measured for each region of interest and normalized to the area. All measurements were taken with the same exposure times and immunolabeling was also performed at the same time." It was not possible for experimenters to be blinded to genotype since the disruption of the pinceau was so obvious.
- ii. Tissue was prepared from mice maintained at the Weizmann Institute of Science in Rehovot, Israel. The *Cntn1*^{-/-} mice are extremely weak and frequently die before P18. In fact, P18 *Cntn1*^{-/-} mice were the oldest we were able to obtain and then only 2 survived out of many litters. We most frequently had *Cntn1*^{-/-} mice die at P16 or P17 and had to wait for many litters to get P18 mice.
- iii. *Why the difference in number of pinceau between control and Cntn1^{-/-} mice?*
We first quantified the mean Kv1.2 intensity in control mice. As we measured the *Cntn1*^{-/-} mice the difference was so obvious and stark we simply stopped at 55 examples (the statistical comparison also supports this notion). If the editor and reviewer would like to see additional numbers counted we can do this, but it will not change the significance of the results in any way.
5. *Methods for Fig 8d: how was an innervated AIS defined objectively?*
- i. As described in an expanded methods section we describe the following: A PyN AIS is considered innervated when there are at least two ChC boutons from the same cartridge present at the AIS.
6. *Methods for Fig 8f/h: what were the objective criteria for inclusion as a punctum? And as 'co-localised' with AIS label? Was this, and all other analysis, carried out blind?*
- i. In a revised methods a punctum was considered positive if the area of the immuno-positive signal for gephyrin or VGAT fell within the range of ~ 0.1 - $0.5 \mu\text{m}^2$ and at least half of it overlapped with the AnkG or $\beta 4$ spectrin immunosignal that outlines the AIS. Additionally, the fluorescence intensity of the punctum had to be more than two-fold higher than the surrounding background. Gephyrin or VGAT puncta density at the AIS was then calculated by dividing the number of PyN AIS gephyrin or VGAT puncta by the length of the AIS. Representative maximum projection images of PyN AIS GABAergic synapses visualized via gephyrin or VGAT immunostaining were generated using a z-series of 10 images with a depth interval of $0.37 \mu\text{m}$.

- ii. All analyses were performed blinded. This statement was added to the methods.
7. *Figure S1a: please annotate the images of the panel. What is the signal detected here? Is it streptavidin?*
 - a. Done. Yes, the labeling is streptavidin. This has now been added to the figure legend.
 8. *Lines 135 and 136: The sentence 'all experiments were performed three times independently for each developmental timepoint with 2 million neurons per experiment' should be moved to the corresponding figure legend or methods.*
 - a. Done.
 9. *Figure S2: the authors should specify what 'replicates' mean in this context: are these technical replicates (e.g., plates) or biological replicates (cultures)?*
 - a. These are biological replicates -now stated in the methods. Different cultures and experiments performed on different days – weeks apart in fact.
 10. *Lines 176 and 177: The statement 'Similarly, Nav1.2 (encoded by Scn2a) the main voltage-gated Na⁺ channel expressed at the AIS in DIV14 hippocampal neurons' should include a reference to existing literature.*
 - a. I am unaware and unable to find a paper showing this. This statement was based on previous unpublished experiments performed in the Rasband laboratory. There are many papers illustrating that Nav1.2 is the main Na⁺ channel in developing neurons in vivo. Nevertheless we agree with the reviewer that the statement should be justified by literature. We cannot justify this and have therefore modified the statement to reflect the literature. Therefore, we replaced 'in DIV14 hippocampal' with 'developing'. We now reference Boiko, J Neurosci 2003 who first described this switch from Nav1.2 to Nav1.6 at the AIS.
 11. *Figure 5. It is unclear at which stage of development (DIV) the neurons were infected, as well as when the immunostaining was performed. Please specify in the corresponding part of the text/figure legends.*
 - a. Done.
 12. *Line 254 and figure 5: the authors refer to the labelling showed in figure 5f as 'cortical neurons.' Please specify which area and layer of cortex you are showing.*
 - a. We did not keep track of that level of detail. We observed transduced neurons throughout the cortex and all layers. There was no specificity for one region or layer over another.
 13. *Figure legend on Figure 5: explanation for the 'RO' and Tg abbreviations is missing in the figure legend.*
 - a. Done.
 14. *Figure 6: the neuronal stage/DIV of the pictures showing the immunostaining is not stated either in the results nor in the figure legend. This applies for the stage at which the infection was performed.*
 - a. Done.
 15. *Line 289: to match the previous writing format, please change 'loss of both Nfasc and Cntn1' to 'loss of Nfasc or Cntn1'.*
 - a. Done.
 16. *Figure 7c: please indicate in the figure legend what do the different lines in the violin plots indicate (e.g., mean).*
 - a. Done.
 17. *Line 317: replace 'remains' by 'remain'.*
 - a. Done
 18. *Figure S5c: the results of the paper show that the deletion of Cntn1 leads to 1) a decrease in the percentage of pyramidal neurons innervated by chandelier cells (figure 8d) and 2) a reduction of inhibitory synapses between the two cell types (figure 8f and 8h). However, in Figure S5c, it seems that only the effect 1) is showed. The schematic would be more accurate if the second scenario is also pictured.*

a. We revised this figure as suggested, it is now a new Figure S6.

19. Corrections:

- a. Line 582: please specify the 'PB' abbreviation. -Done
- b. Line 584: please specify the 'PM' abbreviation. - Done
- c. Line 589: please capitalise 'Vector Labs' accordingly. -Done

Reviewer #2

Major points:

1. *The authors make use of a specific antibody targeting the ectodomain of Nfasc to probe its proximal proteome. Since this approach greatly depends on the specificity of the used antibody, the authors also perform a second labeling experiment in which they target another AIS protein, NrCAM (Fig S1C and S1D). Comparison of both antibodies reveals a significant overlap (Fig S1D). It would be informative to visualize the obtained data for both antibodies in a Venn diagram which visualizes the overlap with regards to specific hits that were obtained using both antibodies. Did the authors filter out proteins that were identified with just one of the two used antibodies? The intersect[ion] of both antibodies represent high confidence AIS proteins. An alternative approach would be to use a different antibody targeting the same protein (Nfasc) or to perform the experiment in cells lacking the primary bait to filter out potential false positive hits.*
 - a. Please note that we previously validated the Nfasc antibody used in our experiments on Nfasc knockout tissues. Please see Amor et al., Elife 2017.
 - b. As the reviewer points out, the comparison shown in Figure S1D is essentially the same thing as a Venn diagram. All common proteins identified in both the Nfasc-BAR and NrCAM-BAR lie along the diagonal line shown. Outliers are also clearly detected. The table with the results is included as supplemental table 1.
 - c. The reviewer's suggestion is a good one, and in fact we did this. We used another extracellular Nfasc antibody – A12/18. This antibody was made in the Rasband lab and recognizes an extracellular domain of rat Nfasc (we gave the antibody to the NIH funded Neuromab monoclonal antibody resource for free distribution). Although this antibody works quite well for immunostaining, it was not as good as the chicken Nfasc used throughout this paper. We obtained a data set with far fewer peptide spectral matches. Because of this, we abandoned the use of this antibody for Nfasc-BAR and instead used antibodies against NrCAM. The reviewer's suggestion that we attempt to perform the same experiment using neurons deficient in Nfasc is also a good one. However, all of our experiments were performed using rat neurons and not mouse neurons. In addition, since Nfasc is also found in non-AIS locations, we worried a total loss of Nfasc wouldn't enrich further for AIS-specific pools of cell surface proteins. To overcome this problem, we instead decided to disrupt the clustering of Nfasc at the AIS (by AnkG knockout), then compare the Nfasc proximity proteome in control neurons and those lacking AIS clustered Nfasc. To remove AIS AnkG, we used an efficient adenoviral shRNA (see Hedstrom et al., JCB 2009). Unfortunately, the infection efficiency of this virus was not high enough to transduce all neurons and lose all AIS Nfasc in a given culture dish; although we performed Nfasc-BAR on these samples we were unable to clearly identify unique AIS cell surface proteins from the differential comparison. We decided to omit these results as they do not contribute to the main finding of the paper, which is the identification of Cntn1 as a new AIS cell surface protein that regulates axo-axonic synapse assembly. If the reviewer or editor would like to see these results I am happy to provide them.
2. *The authors should perform GO enrichment analysis of their hits to confirm that these are strongly enriched for extracellular proteins.*
 - a. Done. This is now added as a new supplemental Figure S4.

3. *The 'DIV4' experiment had only 63 identified hits (Fig. S3). Is this technical or biological? The authors may want to repeat this experiment.*
 - a. We repeated this experiment 3 independent times (biological replicates). Please note, we actually identified 436 proteins at DIV4. However, only 63 of those showed a Log₂ fold change > 2.
4. *Furthermore, the authors should plot all their obtained data using hierarchical clustering*
 - a. Respectfully, we have not done this analysis as we do not believe it adds any conceptual or mechanistic insight into the results presented. If the editor believes this is essential, we can perform the analysis. Nevertheless, our paper is already 10 main figures and 8 supplemental figures and very long.
5. *The authors state that ion channels are unexpectedly absent in their Nfasc cell surface proximal proteome and state this may be due to the small number of extracellular tyrosine residues in these ion channels, which are the substrates of the biotinylation reaction. Have the authors investigated whether these ion channels are efficiently solubilized in their lysates? They used a fairly stringent lysis buffer but perhaps the ion channels still end up in the insoluble pellet after extraction (did they spin down the lysate after lysis, unclear from the M&M)? This can be easily investigated using western blotting for these ion channels in the input lysate used for streptavidin enrichment.*
 - a. The Rasband lab previously performed proximity biotinylation of cytoplasmic AIS proteins using BioID-dependent proximity biotinylation. In those experiments, we used the same solubilization buffer and we identified AIS Na⁺ channels (Scn2a) (see Fig. 2d of Hamdan et al., Nature Communications, 2020). We have extensive experience solubilizing and purifying ion channels (for example, see Stevens et al., *Elife*, 2021; Ogawa et al., *J Neurosci* 2010; Rasband et al., *J Cell Biol*, 2002).
6. *The authors validated 23 hits using CRISPR based tagging and immunofluorescence (figure 4 (new Figure 5)). Were these 23 hits the only proteins the authors tagged and validated? Or did they also tag additional proteins which subsequently could not be validated to localize to the AIS?*
 - a. The 23 candidates we tested in this paper were the first set tested based on the rationale described in the paper. Please note: we constructed 2 AAVs for each target (total of 46 AAVs). In this paper we illustrate the strategy and show that it can be used to identify new AIS proteins (e.g. Cntn1). Since this original test, we have continued to perform CRISPR-based endogenous gene tagging on many candidate AIS proteins identified in the data set found in this paper, and candidates identified in Hamdan et al., 2020, *Nature Communications*. We have not added all of these proteins because we believe it would distract from the main message of the paper – that Cntn1 regulates axo-axonic AIS synapses. Nevertheless, we assure the reviewer that we are working hard on several new AIS proteins that were also identified from our proximity biotinylation experiments. We respectfully suggest that these are outside the scope of this paper as we are still working to validate them as bona fide AIS proteins (it is a lot of work as illustrated in Fig. 6).

Reviewer #3

1. *The volcano plots in Fig. 2 are illegible (minor concern).*
 - a. We apologize for this inconvenience and poor figure design. We revised Figure 2 to separate the volcano plots, the developmental changes in NF186 at the AIS, and the heat map of changing membrane protein levels as a function of age. This allowed us to increase the size of the volcano plots to make them more legible. These two figures are now Figures 2 and 3 in the revised manuscript.
2. *It is not at all certain that the analysis of PSM numbers vs. number of extracellular tyrosine residues is pertinent because a tyrosine could be in the ectodomain and yet not be exposed to solvent. A quick look at the model of Ig1-Ig4 of Cntn1 in AlphaFold revealed that there are quite a few of these residues.*

- a. We agree that many of the tyrosines in any given protein may not be accessible or exposed to solvent. Nevertheless, as we emphasize in the text, this analysis was only intended as an estimate of proximity. This was only intended as a potential way to identify new candidates with the estimate of proximity. For example, even though many of the residues in Cntn1 may be inaccessible it was one of our top hits. Thus, this approach may be an underestimate of the proximity. We freely admit this and repeatedly state this was only used as an estimate to reduce the total number of candidates we tested. We had to start someplace (we had 285 candidates that passed our filtering – we had to reduce the list somehow and this seemed like a reasonable way to do it).
3. *It is probably an overstatement to suggest that Cntn1 is recruited by Nfasc/Nrcam at AIS with the data that is presented (lines 278/9). Although it is true that Nfasc and Nrcam bind to Cntn1 based on recent analyses of interactome data, the authors are not proving that Cntn1 binds to either of those proteins at AIS. One idea would be to repeat the experiments of Fig. 6b using a Cntn1 construct that cannot bind to Nfasc. Residues at the Nfasc/Cntn1 interface that disrupt the interactions between the two proteins can be found in PubMed ID # 36329006.*
 - a. This is an excellent suggestion. We thank the reviewer for pointing out this manuscript which we are embarrassed to say that we had not seen (it was published shortly after we completed the first complete version of this manuscript). We performed the suggested experiment and show loss of Nfasc binding and AIS recruitment of Cntn1 after mutation of the 3 residues in Cntn1 identified in Chataigner et al., Nat Comm 2022. These results are now included in a new Supplementary Figure S7.
4. *The authors suggest that removal of Cntn1 alters the AIS matrix, but really their evidence points to a decrease in the recruitment of Tnr. This conclusion would be strengthened if the authors analyzed the recruitment of brevican as well.*
 - a. We analyzed brevican recruitment and found it was also lost from the AIS. These results are now included in a new Supplementary Figure S6.
5. *It seems that Fig. S5 could be used as a final figure in the main manuscript (minor concern).*
 - a. We now include this figure as the final figure in the main manuscript – Figure 10.

REVIEWERS' COMMENTS

Reviewer #1 (Remarks to the Author):

All concerns now addressed

Reviewer #3 (Remarks to the Author):

The authors have addressed the concerns I raised during the initial review. Specifically, they added data that strengthen their findings regarding the interaction between contactin-1 and neurofascin as well as the disruption of the matrix using brevicin staining. At this stage, I have no concern.